# How do Transformers Learn Implicit Reasoning?

**Jiaran Ye**♠† **Zijun Yao**♠† **Zhidian Huang**♠ **Liangming Pan**♡‡ **Jinxin Liu**♠
**Yushi Bai**♠ **Amy Xin**♠ **Weichuan Liu**◇ **Xiaoyin Che**◇ **Lei Hou**♠‡ **Juanzi Li**♠

♠DCST, BNRist; KIRC, Institute for Artificial Intelligence, Tsinghua University, *China*
♡MOE Key Lab of Computational Linguistics, Peking University, *China*  ◇Siemens AG, *China*
{yejr23, yaozj20}@mails.tsinghua.edu.cn
{houlei, lijuanzi}@tsinghua.edu.cn

## Abstract

Recent work suggests that large language models (LLMs) can perform multi-hop reasoning implicitly—producing correct answers without explicitly verbalizing intermediate steps—but the underlying mechanisms remain poorly understood. In this paper, we study how such implicit reasoning emerges by training transformers from scratch in a controlled symbolic environment. Our analysis reveals a three-stage developmental trajectory: early memorization, followed by in-distribution generalization, and eventually cross-distribution generalization. We find that training with atomic triples is not necessary but accelerates learning, and that second-hop generalization relies on query-level exposure to specific compositional structures. To interpret these behaviors, we introduce two diagnostic tools: cross-query semantic patching, which identifies semantically reusable intermediate representations, and a cosine-based representational lens, which reveals that successful reasoning correlates with the cosine-base clustering in hidden space. This clustering phenomenon in turn provides a coherent explanation for the behavioral dynamics observed across training, linking representational structure to reasoning capability. These findings provide new insights into the interpretability of implicit multi-hop reasoning in LLMs, helping to clarify how complex reasoning processes unfold internally and offering pathways to enhance the transparency of such models.

⭘ https://github.com/Jiaran-Ye/ImplicitReasoning

## 1 Introduction

Large language models (LLMs) demonstrate strong performance on complex, multi-step reasoning tasks [9, 7, 29, 1, 32, 19, 14]. Typically, these reasoning abilities are elicited using chain-of-thought (CoT) prompting, which encourages models to explicitly articulate intermediate reasoning steps [28, 35, 6, 33, 30]. Beyond CoT, recent studies indicate that LLMs can also engage in ***implicit reasoning*** [37, 5, 11, 15], producing correct answers without verbalizing intermediate steps.

While implicit reasoning is widely acknowledged, the internal mechanisms that empower this ability remain unclear. In this paper, we aim to uncover the internal processes of implicit reasoning by examining a concrete, structured scenario: *multi-hop implicit reasoning*, where the model must answer compositional queries (*e.g.*, $(e_1, r_1, r_2) \rightarrow e_3$) by implicitly traversing an intermediate entity $e_2$, without explicitly verbalizing it. A fundamental question in this scenario is that: does the model genuinely conduct step-by-step reasoning internally, or is it merely recalling the answer from its memorized knowledge? Although both behaviors can produce correct outcomes, they reflect

---

†Equal Contribution.
‡Corresponding Author.

39th Conference on Neural Information Processing Systems (NeurIPS 2025).

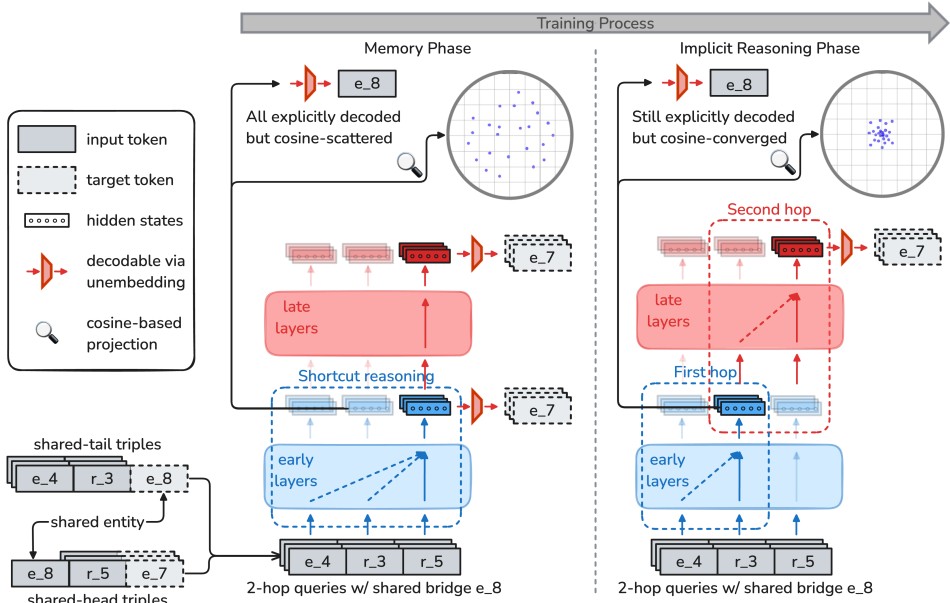

Figure 1: We track the evolution of bridge entity representations across training. In the memory phase (left), intermediate entities are explicitly decodable but geometrically scattered. In the implicit reasoning phase (right), representations converge in cosine space, supporting structured multi-hop reasoning through early-to-late layer transitions. Queries that share an bridge entity are composed for cosine-based representation analysis.

fundamentally distinct cognitive processes. This observation motivates our central research question: *How do LLMs acquire and perform implicit reasoning during training and inference?*

Existing studies that investigate implicit reasoning often rely on pretrained LLMs whose training data lacks precise experimental control, making it challenging to conclusively determine whether models have genuinely learned implicit multi-step reasoning or instead rely on its prior knowledge or shortcut solutions [12, 34, 5]. Symbolic datasets [25, 27, 26] partially alleviate this concern by training models from scratch, yet they still lack the fine-grained experimental control and behavioral granularity necessary for deeper analysis. To address these limitations, we construct *an extended symbolic environment*, featuring targeted omissions and query-level variations, to precisely identify whether implicit reasoning and generalization truly emerge.

To facilitate the analysis under our symbolic environment, we introduce two diagnostic tools that overcome specific limitations of prior methods: (1) *cross-query semantic patching*, which enhances causal interpretability by locating intermediate entity representations based on their semantic transferability across queries rather than solely their impact on final outputs; and (2) *a cosine-based representational lens*, which avoids assumptions inherent in decoding-based probing by examining structural consistency of internal representations across reasoning contexts. Together, these tools enable precise examination of the internal processes driving implicit reasoning.

Our empirical analysis begins with a behavioral study conducted under fine-grained experimental control (Section 2). Under a complete training configuration, we observe that multi-hop implicit reasoning emerges in three distinct stages: memorization, in-distribution generalization, and finally cross-distribution generalization. Through ablation studies, we further demonstrate that while exposure to in-distribution (ID) triples is not strictly necessary for achieving in-distribution generalization, its absence significantly delays the onset of this behavior. Additionally, we find that generalization to second-hop queries fails unless the model encounters exact compositional structures during training, revealing a strong dependency on query-level exposure.

These behavioral insights reveal previously unreported patterns, motivating us to revisit and probe the internal mechanisms of implicit reasoning. In Section 3, we first use cross-query semantic patching to localize intermediate entity representations, typically identifying them within the middle layers

corresponding to the $r_1$ tokens. We then test the common assumption that intermediate entities are explicitly decodable from internal states and find this assumption inconsistent with our observed reasoning behavior. This disconnect leads us to adopt a geometric perspective, wherein successful reasoning strongly correlates with consistent clustering of intermediate representations within cosine similarity space (Figure 1).

In Section 4, we close the loop by explicitly connecting these internal representational mechanisms to external behavioral patterns. We demonstrate that successful generalization robustly correlates with the clustering structure of intermediate representations across diverse queries and training distributions. Although in-distribution (ID) triple supervision is not required to induce this clustering, it substantially accelerates its emergence by constraining the representational space early in training. Finally, we identify that what appears to be first-hop generalization to out-of-distribution (OOD) triples is actually an artifact arising from representational alignments induced by ID exposure, highlighting the fragile and data-dependent nature of implicit generalization.

Collectively, our results provide a comprehensive account of how implicit multi-hop reasoning emerges within LLMs — grounded in observable behaviors, elucidated through mechanistic analyses, and offering foundational insights for future studies on model interpretability.

## 2 Behavioral Signatures of Implicit Reasoning under Fine-Grained Control

Existing studies on implicit reasoning fall into two broad categories, each with notable limitations. (1) Analyses based on pretrained LLMs operate in an uncontrolled setting where the training data are opaque—making it difficult to distinguish genuine reasoning from memorization. (2) In contrast, recent works adopt symbolic datasets with synthetic training from scratch [25], but primarily focus on *dataset-level trends*, without isolating *what specific training signals are necessary* for solving each compositional query.

We argue that an ideal analysis setting should satisfy three key properties: (1) **compositional structure**, to support multi-step inference; (2) **fine-grained control**, to support query-level ablations and conditionally constructed variants; and (3) **behavioral resolution**, to distinguish between memorization, generalization, and reasoning. With these goals in mind, we construct a symbolic training environment that extends prior datasets with new configurations and targeted omissions, and reveals several behavioral phenomena not captured in prior work.

To achieve this, we adopt GPT-2 as our base model due to its balance of capacity and tractability, and verify the scalability of results using larger models. Full training details are provided in Appendix G.

### 2.1 Data Construction: Fine-Grained Control for Compositional Reasoning

To enable fine-grained behavioral analysis, we extend the symbolic reasoning setup of Wang et al. [25] with expressive query-level control configurations. The data comprises atomic triples and compositional queries:

- **Atomic Triples.** Each atomic fact is represented as a triple $(e_1, r_1) \rightarrow e_2$. This formulation mimics simple factual relations such as *(Alice, mother-of)* $\rightarrow$ *Beth* and *(Beth, sister-of)* $\rightarrow$ *Carol*, serving as the atomic unit of the reasoning environment. The triples are partitioned into two subsets: *In-Distribution (ID) Triples* are used in both standalone form and as components of multi-hop training queries; *Out-of-Distribution (OOD) Triples* appear in training data only in standalone form, and are excluded from multi-hop composition, enabling the creation of test queries involving out-of-distribution reasoning. Note that ID and OOD triples share the same set of entities and relations.

- **2-Hop Queries.** Each reasoning task takes the form of a compositional chain $(e_1, r_1, r_2) \rightarrow e_3$, where the model performs implicit reasoning over an bridge entity $e_2$. For instance, the model receives only the compositional query *(Alice, mother-of, sister-of)* and is expected to predict the correct target *Carol*, implicitly reasoning through the intermediate entity *Beth*. We distinguish: *Test-OI*: test queries where the first hop comes from an OOD triple and the second hop from an ID triple; *Train-II*: queries with both hops from ID triples used during training. Other query types, such as *Test-II*, *Test-IO*, and *Test-OO*, follow similar definitions.

**Training Configurations.** Our **base configuration** (Figure 6a) includes all atomic triples (ID and OOD) and the full set of Train-II queries, and evaluate on Test-II, Test-OI, Test-OO, and Test-IO, allowing comprehensive generalization assessment. To isolate the conditions for generalization, we define **a flexible family of training variants** that omit specific triples, restrict compositional roles, or remove entire subsets, allowing targeted query-level ablations. This design supports controlled investigations into the functional dependencies behind implicit reasoning behaviors.

**Extension to 3-hop Reasoning.** Although our main analyses focus on 2-hop queries, the same construction framework naturally applies to 3-hop settings, exhibiting consistent behavioral and mechanistic patterns. For more details, refer to Appendix C.

Together, these configurations serve as the foundation for our study. Further dataset construction details and illustrations are available in Appendix B.

## 2.2 Three-Stage Generalization

Leveraging the base configuration introduced in Section 2.1, we track model performance throughout training and observe a striking behavioral trajectory that unfolds in three distinct phases (Figure 2):

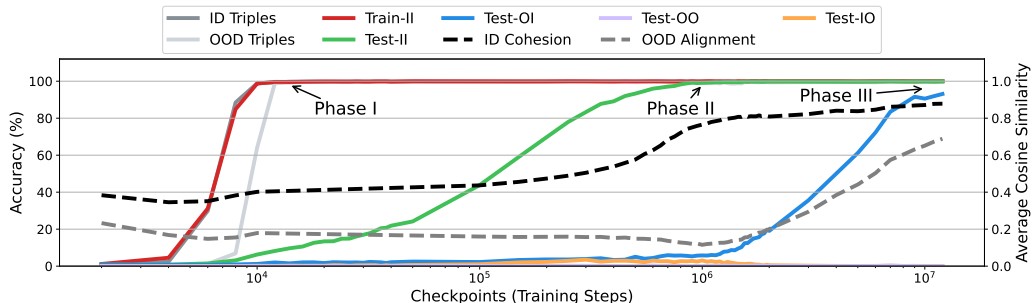

Figure 2: Training dynamics under the base configuration, revealing three distinct phases. Accuracy curves track model performance on different query types, while dashed lines plot the ID Cohesion and OOD Alignment scores (Section 3.3).

**Phase I: Memorization.** The initial stage involves quickly fitting the training data, including atomic facts and 2-hop compositions. The model memorizes these facts, but generalization to unseen queries remains minimal.

**Phase II: ID Generalization.** After memorization saturates, the model begins to generalize to Test-II queries (unseen ID-ID compositions), marking a shift from memorization to compositional generalization within ID, akin to the *grokking* phenomenon described by Wang et al. [25].

**Phase III: Cross-Distribution Reasoning.** The model next learns to generalize across distributions, gradually incorporating OOD triples in the first hop while maintaining the ID in the second. This transition is slower than Phase II and requires more training. Building on the grokking phenomenon, our analysis uncovers this additional phase of generalization across distributional boundaries.

Interestingly, generalization fails consistently when the second hop is from OOD triples, revealing a stronger bottleneck in the second relational step. These phases show that reasoning develops in structured stages, each with distinct patterns of success and failure, highlighting the need to treat reasoning not as a monolithic ability, but as a set of behaviors with separable developmental conditions.

## 2.3 ID Triples Are Not Required for ID Generalization—but Accelerate It

Prior work has repeatedly observed that while models can correctly answer individual atomic triples, they often fail to generalize to 2-hop queries constructed by composing those same triples[3, 31, 42]. Additionally, in Section 2.2, we observe that our model quickly memorized atomic triples (Phase I) but took longer to generalize to Test-II queries. These findings raise a natural question: *Are atomic ID Triples actually necessary for learning ID-based 2-hop reasoning?*

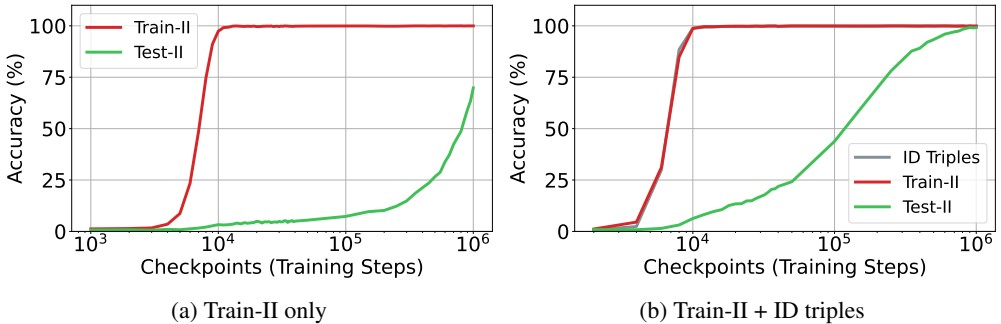

(a) Train-II only            (b) Train-II + ID triples

Figure 3: Training trajectories under two configurations used in Section 2.3. (a) Only Train-II queries are used, with no exposure to atomic ID triples. (b) Both Train-II queries and atomic ID triples are included in training.

To investigate this, we test a minimal training configuration that excludes both ID and OOD Triples, using only Train-II queries. Surprisingly, the model still generalizes to unseen ID combinations Test-II (Figure 3a). That is, training with Train-II alone is sufficient for ID-based generalization.

However, when comparing this minimalist setting to a variant where ID Triples are included[1] alongside Train-II, we found that generalization to Test-II occurs significantly earlier (Figure 3b), suggesting that while atomic facts are not required for generalization, they **accelerate** learning.

### 2.4 Second-Hop Generalization Requires Query-Level Training Match

While the model can generalize when OOD triples appear in the first hop (Section 2.2), it consistently fails when the OOD component is in the second hop. This raises a natural follow-up question: *What training data is necessary to enable second-hop generalization within the ID domain*?

Building on Section 2.3, where training with only Train-II queries still led to ID generalization, we perform a targeted ablation to isolate the role of second-hop coverage. Specifically, we remove a subset of atomic triples (e.g., $(e_B, r_5, e_F)$) from being used as second hops in any Train-II query. We then test whether the model could correctly answer Test-II queries that involved these **excluded triples as second-hop** (Figure 6b).

We find that the model consistently fails to answer these Test-II queries, while performance on other queries remained unaffected, even when the same atomic triples were used in the first hop. This confirms that **second-hop generalization requires query-level training match**: the model must encounter the specific second-hop composition during training to generalize over it. Exposure to the same facts in other structural roles is not sufficient.

To further validate this finding, we replicate the ablation under the full base configuration (Appendix D.1) and observe the same failure. Additionally, we analyze how second-hop exposure frequency impacts query acquisition order (Appendix D.2). We found that Test-II queries involving a particular atomic triple as the second hop were answered correctly earlier when that atomic triple appeared more frequently as a second hop during training.

## 3 Locating and Characterizing Reasoning Representations

In Section 2, we observed surprising behavioral phenomena that offer new insights into implicit reasoning in Transformers. These findings challenge prevailing assumptions, such as the necessity of exposure to atomic facts [40, 36] or the impossibility of OOD generalization [25].

To better understand these behaviors, we shift our focus from **what the model does** to **how the model achieves it internally**. Specifically, we examine the intermediate entity that connects the two relational steps. Any correct solution, at least implicitly, passes through this latent bridge, making it a key target for probing the model's reasoning process. To this end, we first introduce the causal

---

[1]We exclude OOD Triples from both configurations to ensure comparability: since the "Train-II only" configurations contains no OOD facts, generalization to Test-OI is inherently impossible.

probing method *Cross-Query Semantic Patching* (Section 3.1) to locate these intermediate entities in representations. We then revisit whether internal states are decodable using logit lens (Section 3.2). We finally explore how geometric regularity in these representations supports the model's ability to generalize across queries (Section 3.3).

### 3.1 Locating Intermediate Entity Representations via Cross-Query Patching

To analyze how transformers internally represent intermediate reasoning steps, a crucial first step is to identify **where** such representations are encoded in the model's hidden states.

Existing methods such as linear probing and causal patching offer only partial insight. Linear probing reveals correlations between hidden states and output tokens, but not their causal role in reasoning. Causal patching assesses causal influence, typically measures whether a random source activation affects the target's output, but doesn't assess what the activation semantically represents [25].

**Cross-Query Semantic Patching.** To go beyond correlation or superficial causal influence, we introduce *cross-query semantic patching*, a method designed to test whether a hidden representation encodes a semantically valid intermediate entity. Specifically, given a source query $(e_1, r_1, r_2)$, we test a set of candidate hidden states from different layers and positions (*e.g.*, layer 3 at the $r_1$ position) that may contain the bridge entity representation. For each candidate, we insert its hidden vector into a structurally similar target query $(e_5, r_6, r_7)$ at the same position, replacing the original hidden state.

If the patched model's prediction changes from the original reasoning path $r_7(r_6(e_5))$ to $r_7(r_1(e_1))$, this indicates that the inserted representation carries transferable semantic information corresponding to the bridge entity.

We apply this patching procedure across multiple layers and token positions, with three settings that differ in both the source of the intermediate entity and the model's training stage: (1) Phase II with ID-derived intermediate entities, (2) Phase III with ID-derived intermediate entities, and (3) Phase III with OOD-derived intermediate entities. This alignment ensures that patching is conducted under conditions where the model is capable of reasoning over the relevant intermediate entity type. For completeness, we also report Phase I results, where patching yields negligible success, confirming that reasoning-relevant representations emerge only after generalization.

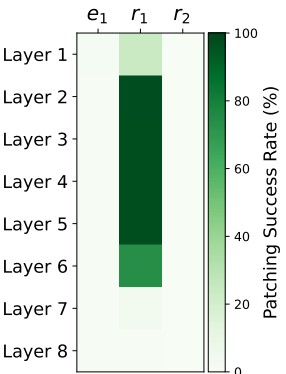

Figure 4: Average patching success rate across layers and token positions.

Detailed per-setting results are presented in Appendix F. We report the average patching success rate across these three settings in Figure 4. This aggregated result shows that effective patching occurs primarily at the $r_1$ token position in the middle layers. In the following analyses, we use **layer 5 at the $r_1$ token position** of our 8-layer GPT-2 model as the reference point, denoting the hidden state as $\mathbf{h}_{r_1}^5$.

### 3.2 Explicitly Decodable ≠ Implicitly Informative

Having located the positions encoding intermediate entities, we next ask whether their internal role can be explained using existing interpretability tools. A key assumption in prior work is that if a hidden state encodes an intermediate entity, it should be decodable into a human-interpretable token, for example via the logit lens [18, 16, 33, 22, 40]. We test this assumption by measuring the decodability of $\mathbf{h}_{r_1}^5$, and examining whether decodability aligns with the emergence of reasoning behavior across training phases.

**Setup.** We adopt the logit lens to evaluate decoding performance in two modes: (1) **Immediate probing**: projecting the extracted hidden state onto the output vocabulary directly. (2) **Full-run probing**: patching the extracted hidden state into a randomly selected query at the corresponding position, and decoding it after processing through the model's layers. These methods assess whether the hidden state contains a token-level signal or whether the model itself can internalize and recognize

Table 1: Success rates (%) of explicitly decoding intermediate entities from $\mathbf{h}_{\mathbf{r}_1}^5$ across reasoning phases and decoding methods.

| Source | Immediate Probing | | | Full-run Probing | | |
|--------|---------|----------|-----------|---------|----------|-----------|
| | Phase I | Phase II | Phase III | Phase I | Phase II | Phase III |
| ID-derived | 92.1 | 98.8 | 99.9 | 97.1 | 99.9 | 99.9 |
| OOD-derived | 67.7 | 81.3 | 99.8 | 83.7 | 98.6 | 99.7 |

it. For each phase, we compute the decoding success rate for intermediate entities grouped by their origin—either from ID or OOD triples[2].

**Result 1: Decodability does not correlate with reasoning emergence.** As shown in Table 1, decoding success remains high and stable for ID-derived representations across all phases. However, implicit reasoning capabilities only emerge after Phase II, suggesting that decodability alone does not explain reasoning emergence.

**Result 2: No decodability gap between ID and OOD sources during cross-distribution generalization.** In Phase II, while the model generalizes to ID-ID (Test-II) queries but fails on ID-OOD (Test-OI) queries, there is no significant difference in decoding success between ID-derived and OOD-derived representations. This further demonstrates that representations can be equally decodable yet differ in whether they are functionally recognized and utilized by the model.

**Implication.** These results indicate that explicit decodability alone cannot explain when or how a representation contributes to reasoning. Even when a representation can be decoded correctly, the model may not rely on it for reasoning.

To further probe the role of explicit decoding in reasoning, we constructed a controlled setting where the model was incentivized to represent intermediate entities in a decodable form. Interestingly, the model initially attempts this strategy but quickly abandons it, indicating that the model prefers non-explicit representations for generalization. We provide details of this experiment in Appendix E.

## 3.3 Geometric Regularity of Intermediate Representations via Cosine Lens

The gap between explicit decodability and actual usage motivates a different approach. Instead of asking *"Can we decode what this hidden state?"*, we ask *"How is this representation organized across different contexts?"*

Most prior work focuses on decoding representations, but we take the reverse approach: **given a known intermediate entity**, can we identify recurring structure in how the model represents it across contexts to achieve consistent representations [24]? This reverse mapping is enabled by our earlier analysis in Section 3.1, which identifies the position $\mathbf{h}_{\mathbf{r}_1}^5$ encoding the intermediate entity. At this anchor, we collect hidden states from queries sharing the same intermediate entity (Figure 1), and examine whether these vectors reflect a consistent internal pattern.

To assess consistency, we focus on structural alignment in the model's embedding space, using *cosine similarity*—a common metric for semantic proximity in high-dimensional representations[8, 21, 13]. This allows us to examine whether the model reuses internal abstractions through representational geometry, instead of relying on explicit decodability.

**Case Study: Visualizing Representational Clustering.** To gain an initial sense of the representational patterns that emerge during training, we visualize the hidden states of a randomly selected intermediate entity that appears in multiple two-hop queries. For this entity, we extract $\mathbf{h}_{\mathbf{r}_1}^5$ across relevant queries instances (where the intermediate entity is either ID-derived or OOD-derived) and compute pairwise cosine distances (defined as $1 - $ cosine similarity) between them. We then project these high-dimensional vectors into two dimensions using *Multidimensional Scaling (MDS)*

As shown in Figure 5, hidden states for a common intermediate entity form distinct geometric patterns across the phases. In Phase I, both ID-derived and OOD-derived representations are scattered; in Phase II, ID-derived representations form tight cosine-space clusters, marking the transition to

---

[2]Notably, the origin of an intermediate entity depends solely on the first hop and is independent of the second-hop configuration.

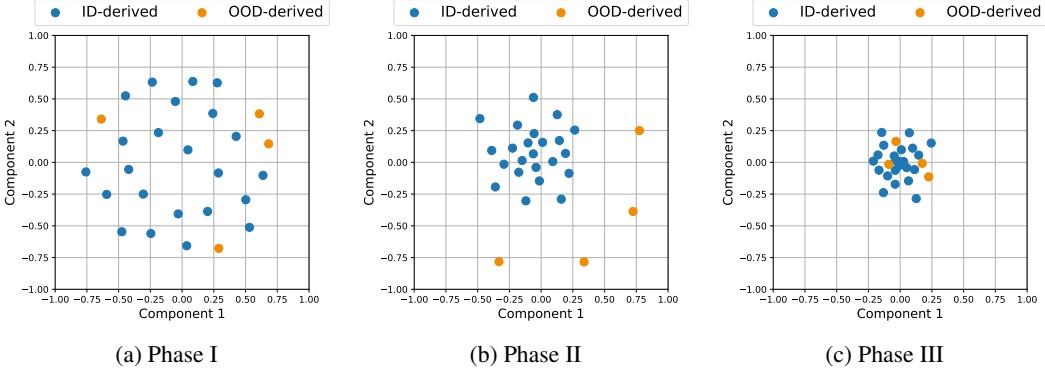

| (a) Phase I | (b) Phase II | (c) Phase III |

Figure 5: Cosine-space projection of a random intermediate entity across three training phases.

in-distribution reasoning; In Phase III, OOD-derived representations also begin to align with the ID-based cluster, signaling cross-distribution generalization. This suggests that latent variables are reused not by explicit decoding, but through the emergence of a consistent geometric structure.

**Quantifying Representational Convergence.** Encouraged by this observation, we next quantify the consistency of entity-level representations across the full dataset. We define two metrics: (1) **ID Cohesion Score:** the average cosine similarity between ID-derived representations and their centroid, reflecting in-distribution consistency. (2) **OOD Alignment Score:** the average cosine similarity between OOD-derived representations and the same ID centroid, reflecting how well the model unifies cross-distribution representations. These scores are computed on a per-entity basis and averaged across all intermediate entities.

Tracking these scores, we find that the **ID Cohesion Score** rises steadily, aligning with Test-II generalization, while the **OOD Alignment Score** starts increasing later, following the rise in Test-OI performance (Figure 2). This suggests that successful implicit reasoning relies on representational consistency across diverse contexts: only when hidden states for the same entity align closely in cosine space can they be consistently reused for multi-hop inference. Thus, **cosine-space clustering** emerges as the model's internal mechanism for semantic abstraction and generalization.

## 4 Closing the Loop: Explaining Behavioral Phenomena

Drawing on the mechanistic evidence in Section 3.3, we revisit our empirical observations in Section 2 and explain how they emerges from the internal dynamics of representation formation.

### 4.1 Clustering of OOD-Derived Representations Driven by ID Supervision

In Phase III, where the model successfully performs OOD reasoning at the first hop, the clustering of OOD-derived representations in cosine space plays a crucial role. However, while the clustering of ID-derived representations is expected due to direct supervision from Train-II queries, the alignment of OOD-derived representations with ID clusters is less intuitive. OOD-derived entities are never explicitly supervised as intermediate steps in multi-hop queries, making their eventual clustering with ID entities surprising.

We hypothesize that **frequent exposure to atomic triples** is the driving factor behind the observed alignment. This exposure leads the model to assimilate the OOD representations into the existing ID clusters, thereby stabilizing them. To validate this hypothesis, we designed an ablation study where we varied the ID/OOD ratio across three configurations: 0.8/0.2, 0.5/0.5, and 0.3/0.7. Only the 0.8/0.2 ratio demonstrates successful generalization into Phase III, effectively answering Test-OI queries, while the others failed to achieve this stage. Detailed results are provided in Appendix H.

### 4.2 ID Triple Queries Accelerate Generalization by Constraining the Representation Space

Knowing that generalization relies on the clustering of intermediate representations, we link the acceleration effect (Section 2.3) brought by ID triple to the property of autoregressive Transformers:

both ID triple queries $(e_1, r_1)$ and two-hop queries $(e_1, r_1, r_2)$ produce **the same hidden state at the $r_1$ token position** due to causal masking, where the model only attends to tokens preceding $r_1$.

Since our mechanistic analyses anchor at $r_1$ position ($\mathbf{h}_{r_1}^5$), the hidden states we study are indistinguishable across both query formats. This means ID triple supervision directly shapes the same representations used in multi-hop reasoning. While the ID triple task optimizes for explicit decoding—mapping $(e_1, r_1)$ to $e_2$—it doesn't guarantee functional reasoning (Section 3.2). However, it plays a crucial role by **constraining** the $r_1$ hidden state to lie within a subspace that supports entity decoding, thereby limiting the model's search space during generalization to a smaller region.

To validate this mechanism, we construct an ablation configuration that *removes* a subset of ID triple (*e.g.*, $(e_A, r_1, e_B)$) from training, while still including their corresponding two-hop compositions (*e.g.*, $(e_A, r_1, r_2)$) in Train-II. We then test whether the model can recover the held-out ID triples at test time (*e.g.*, given input $(e_A, r_1)$, predict $e_B$). The results align with expectations: the model is able to correctly predict $e_B$ (see Appendix I for details). This outcome demonstrates that the $r_1$ hidden state associated with $(e_A, r_1)$ lies in the same region as other Train-II queries sharing latent $(e_B)$—a region already shaped by remaining ID triples involving $e_B$ (*e.g.*, $(e_X, r_7, e_B)$):

$$\underbrace{(e_A, r_1)}_{\text{held-out triple query}} \xrightarrow{\text{share } r_1 \text{ position with}} \underbrace{\left\{ \begin{array}{l} (e_A, r_1, (e_B), r_2) \\ (e_X, r_7, (e_B), r_3) \end{array} \right\}}_{\text{Train-II with latent } (e_B)} \xrightarrow{\text{constrained by}} \underbrace{(e_X, r_7)}_{\text{retained ID supervision}} \xrightarrow{\text{decodable}} e_B$$

This supports the claim that ID triple supervision constrains the $r_1$ hidden state to a decodable region, facilitating clustering. Consequently, as the model enters Phase II and learns two-hop reasoning, it refines representations within an already structured subspace, speeding up representational convergence and behavioral generalization compared to a without ID configuration where clustering must be learned from scratch.

## 4.3 Why the First Hop, and Only the First Hop, Generalizes to OOD?

The intended supervision signal from Train-II queries is to teach the model a structured two-step reasoning process: $\text{token}(e_1) \xrightarrow{r_1^{\text{2hop}}} \text{latent}(e_2) \xrightarrow{r_2^{\text{2hop}}} \text{token}(e_3)$, where both $r_1^{\text{2hop}}$ and $r_2^{\text{2hop}}$ are relations applied in a purely compositional context, independent of atomic triple learning (Section 2.3).

In contrast, atomic triples expose the model to direct mappings of the form: $\text{token}(e_1) \xrightarrow{r_1^{\text{atomic}}} \text{token}(e_2)$, which train a shallow predictive behavior over observed fact pairs.

It is therefore surprising that models can correctly answer Test-OI queries, suggesting that the mapping $r_1^{\text{atomic}}$ somehow transfers into $r_1^{\text{2hop}}$, allowing the model to reuse OOD-derived $\text{token}(e_2)$ representations for implicit reasoning. However, this apparent generalization is in fact a side-effect of representational alignment induced by ID triples.

As established in Section 4.2, the hidden state at the $r_1$ token position is shared across atomic and 2-hop queries, encouraging the model to align $\text{latent}(e_2)$ with $\text{token}(e_2)$ for ID triples. Separately, as shown in Section 4.1, the model gradually pulls OOD-derived representations into the same cosine-space cluster as ID-derived representations. These two mechanisms together enables OOD-derived $\text{latent}(e_2)$ to match the expected format of $r_1^{\text{2hop}}$:

$$\text{OOD-derived token}(e_2) \xrightarrow{pulled\ by} \text{ID-derived token}(e_2) \xrightarrow{align\ with} \text{latent}(e_2),$$

In this sense, the model doesn't truly generalize OOD $r_1^{\text{atomic}}$ into $r_1^{\text{2hop}}$—it "cheats" by reusing shared representation scaffolds shaped by ID triples. The success on Test-OI queries is thus illusory: what appears to be a generalization is in fact a misalignment between model structure and supervision.

Viewed from this perspective, the failure of second-hop generalization is not an exception. Unlike the first hop, the model cannot rely on representational anchoring from shared prefixes and must learn behavior through direct query-level supervision. In contrast, first-hop OOD generalization is an exception, made possible by incidental alignment from overlapping input contexts, hence this does not extend to deeper reasoning. A similar pattern holds in 3-hop reasoning: generalization is only

observed when the reasoning path beyond the first hop remains within ID (Test-III and Test-OII), highlighting the need for explicit supervision in later steps (Appendix C).

We hypothesize that without the representational anchoring effect induced by ID supervision, OOD triples fail to form functionally useful intermediate representations. To test this, we construct a configuration with only OOD triples and Train-II queries, removing ID triples. In this setup, the model fails on Test-OI generalization, confirming that in the absence of ID-based anchoring, OOD triples alone cannot support implicit multi-hop reasoning. See Appendix J for details.

## 5 Discussion

Our study, conducted in a controlled symbolic dataset environment, reveals key insights into the mechanisms of implicit reasoning in transformers, highlighting specific patterns and behaviors that clarify how multi-hop implicit reasoning emerges. These findings may provide valuable answers to existing questions about the implicit reasoning capabilities of LLMs. For instance, our observation regarding the requirement for query-level match offers a potential explanation for why knowledge learned from single-hop tasks does not easily transfer to multi-hop reasoning in LLMs [3, 39, 31]. However, it is important to note that LLMs operate with far richer and more complex knowledge bases, and their internal knowledge interaction mechanisms likely differ from those in our controlled environment. Therefore, while our findings offer useful insights, they should be regarded as preliminary guidance rather than a complete explanation of the reasoning dynamics in LLMs.

## Acknowledgement

This work is supported by Beijing Natural Science Foundation (L243006), National Natural Science Foundation of China (62476150), and the Tsinghua University (Department of Computer Science and Technology)-Siemens Ltd., China Joint Research Center for Industrial Intelligence and Internet of Things (JCIIOT).

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

# A    Related Work

**Mechanistic Exploration of Implicit Reasoning.**    The initial focus on the mechanisms of implicit reasoning arose from the discovery that traditional single-hop knowledge editing methods are ineffective in the context of multi-hop implicit reasoning [41, 38, 12]. This phenomenon since gained more significant attention, prompting researchers to investigate the mechanisms underlying implicit reasoning. Some works suggest that the failure of implicit reasoning is due to the intermediate entities not being properly processed [22, 33]; Other works suggest that the reason lies in the intermediate entities being processed, but not being passed to the correct position [16, 36, 5]. The parallel exploration paths within the model are also a research direction [23, 27]. Recent works have found that the different knowledge composed as different hops used in implicit reasoning is stored in different layers of the model [40, 36]. Additionally, some studies propose that models rely on shortcuts to successfully complete implicit reasoning [34].

**Conditions for Learning Implicit Reasoning.**    Some works have found that the conditions for models to learn implicit reasoning are quite demanding. Some have discovered through custom symbolic datasets that models generalize implicit reasoning abilities only after grokking, and that they require highly compositional data [25]. Other works have found that simply providing optical knowledge is insufficient for enabling models to perform implicit reasoning, it requires training with corresponding multi-hop reasoning data [31, 39, 42]. Moreover, it has been observed that when the two pieces of foundational knowledge used for 2-hop implicit reasoning appear in different paragraphs of the training corpus, the model struggles to combine them for implicit reasoning. However, when these pieces appear within the same paragraph, the model's accuracy in answering the corresponding 2-hop queries significantly improves [3].

**Probing Intermediate Entities.**    As for the probing tools, most of the work uses decoding-based methods to probe intermediate entities. Many works assume that if a model encodes an intermediate entity, it should be able to explicitly decode it. They use the logit lens [18]. as evidence of the presence of intermediate entities in implicit reasoning [22, 33, 16]. A few studies have also trained a linear transformation layer to decode intermediate entities [23].

**Latent CoT.**    Although current Chain-of-Thought (CoT) based reasoning models have shown impressive performance, some perspectives argue that the thought chains do not truly reflect the model's reasoning process [17, 2]. Some works have begun to explore Chain of Thought (CoT) methods that do not verbalize intermediate steps. One approach is to replace the Chain of Thought (CoT) with dots, adjusting the number of dots based on the original length of the thought chain [20, 4]. Another approach to avoid verbalizing is to bypass decoding the tokens in the CoT process and directly use the last hidden state from the previous step as the input vector for the next step [10]. These approaches have achieved results comparable to traditional CoT on specific tasks.

# B    Dataset Details

Our data construction pipeline is adapted from the open-source code released by Wang et al. [25], with modifications to support fine-grained query-level control.

**Entity and Relation Vocabulary.**    In all configurations, we construct a symbolic environment consisting of 2000 entities and 200 relations, each assigned a unique token with no inherent semantics. Since the model is trained from scratch, it has no prior knowledge about the symbols, and learning depends entirely on compositional supervision.

**Atomic Triple Generation.**    We generate 40000 atomic triples of the form $(e_1, r_1) \rightarrow e_2$, where each entity $e_1$ is randomly assigned 20 outgoing relations, and for each such relation $r_1$, the tail entity $e_2$ is sampled uniformly from all entities. These triples are then randomly partitioned at the triple level into ID and OOD subsets, with a default OOD ratio of 5%, while entities and relations remain globally shared across all subsets.

**2-Hop Query Construction.**    Two-hop queries are compositional chains of the form $(e_1, r_1, r_2) \rightarrow e_3$, where the model must implicitly reason through an intermediate entity $e_2$. These queries are

constructed by pairing atomic triples: if both $(e_1, r_1, e_2)$ and $(e_2, r_2, e_3)$ exist, the corresponding query is eligible. Training queries (*Train-II*) are sampled from the set of ID–ID chains. The remaining queries are categorized as *Test-II*, *Test-IO*, *Test-OI*, or *Test-OO* depending on whether the first and second hops are ID or OOD.

We construct the training set with a 7.2:1 ratio of Train-II queries to ID atomic triples to ensure compositional supervision dominates, and sample a fixed test set of 3,000 examples for each type. Table 2 summarizes the key dataset statistics.

**Illustrations of Data Construction Configurations.** To aid in understanding the dataset construction process, we provide two representative configurations as illustrations (Figure 6).

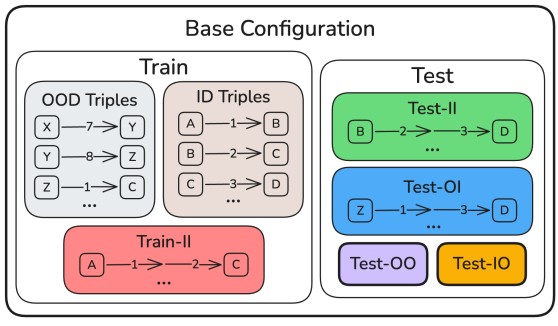
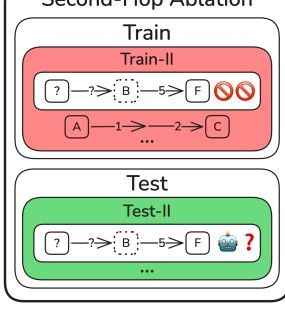

(a) Base configuration                                                     (b) Second-hop ablation

Figure 6: Two representative data construction configurations. **(a) Base configuration:** Contains all atomic triples and Train-II queries, with evaluation covering all test query types. **(b) Second-hop ablation configuration:** A targeted setup where a subset of atomic triples (e.g., $(e_B, r_5, e_F)$) are excluded from appearing as second hops in any training query, while corresponding second-hop queries remain present in the test set.

Table 2: Dataset Statistics

| Data Type | Split | Count |
|---|---|---|
| Vocabulary | Entities | 2000 |
| | Relations | 200 |
| Atomic Triples | In-Distribution (ID) | 38000 |
| | Out-of-Distribution (OOD) | 2000 |
| 2-hop Queries | Train-II (ID $\rightarrow$ ID) | 273600 |
| | Test-II (ID $\rightarrow$ ID) | 3,000 |
| | Test-IO (ID $\rightarrow$ OOD) | 3,000 |
| | Test-OI (OOD $\rightarrow$ ID) | 3,000 |
| | Test-OO (OOD $\rightarrow$ OOD) | 3,000 |

**Design Choices and Assumptions.**

In designing our dataset, we made two key parameter choices to support our analysis focus and the intended analogy to real-world LLM behavior:

First, we fix a relatively high ratio of **Train-II to ID Triples** (7.2:1), ensuring that compositional supervision dominates over direct fact learning. While prior work [25] has shown that a high Train-II / ID ratio is necessary for generalization on Test-II queries, we do not treat this ratio as a variable of interest. Instead, we maintain it at a sufficient level to ensure the emergence of in-distribution reasoning, and shift our focus to more challenging phenomena such as cross-distribution generalization and the underlying mechanisms that support generalizations.

Second, we choose a high ratio of **ID to OOD atomic triples** (95% ID), which is critical for the representational alignment effects discussed in Section 3 and further validated in Appendix H. In

particular, Phase III generalization relies on OOD-derived intermediate representations aligning with the subspace formed by ID-derived ones—an effect that only arises when ID triples are sufficiently dominant.

We acknowledge that these high-ratio settings may appear biased. However, they reflect plausible properties of real-world pretraining corpora: (1) compositional chains (analogous to our Train-II queries) are likely more common than isolated fact triples; and (2) most knowledge entities are learned in richly connected contexts (analogous to our ID triples), while only a minority are sparsely anchored or appear in isolation (analogous to our OOD triples). From this perspective, our symbolic setup not only enables controlled analysis, but also approximates realistic data imbalance patterns observed in LLM pretraining. Accordingly, we treat both ratios as default conditions rather than ablation variables.

## C   Extension to 3-Hop Reasoning

We extend our symbolic framework to support 3-hop queries and verify that the behavioral and mechanistic patterns observed in the 2-hop setting also emerge in deeper compositional regimes.

### C.1   3-hop Dataset Construction

The overall construction methodology mirrors that of 2-hop queries, as described in Appendix B, with an additional relational step. These queries take the form $(e_1, r_1, r_2, r_3) \rightarrow e_4$, where the model must implicitly traverse three relational steps through two intermediate entities.

However, the increased compositional depth introduces new data balancing challenges. In particular, the combinatorial nature of 3-hop chaining (i.e., atomic composition scales cubically) leads to test set sparsity(especially Test-OOO) if the OOD triple proportion is too small. To mitigate this, we reduce the vocabulary size to 1000 entities and 100 relations, and increase the OOD ratio to 20%, while still maintaining a majority of ID supervision discussed in Appendix B. These adjustments ensure sufficient coverage across all evaluation regimes, including out-of-distribution settings. Table 3 summarizes the key statistics of 3-hop dataset.

Table 3: 3-hop Dataset Statistics

| Data Type | Split | Count |
|---|---|---|
| Vocabulary | Entities | 1000 |
| | Relations | 100 |
| Atomic Triples | In-Distribution (ID) | 8000 |
| | Out-of-Distribution (OOD) | 2000 |
| 3-hop Queries | Train-III (ID $\rightarrow$ ID $\rightarrow$ ID) | 120000 |
| | Test-III (ID $\rightarrow$ ID $\rightarrow$ ID) | 1,000 |
| | Test-IIO (ID $\rightarrow$ ID $\rightarrow$ OOD) | 1,000 |
| | Test-IOI (ID $\rightarrow$ OOD $\rightarrow$ ID) | 1,000 |
| | Test-IOO (ID $\rightarrow$ OOD $\rightarrow$ OOD) | 1,000 |
| | Test-OII (OOD $\rightarrow$ ID $\rightarrow$ ID) | 1,000 |
| | Test-OIO (OOD $\rightarrow$ ID $\rightarrow$ OOD) | 1,000 |
| | Test-OOI (OOD $\rightarrow$ OOD $\rightarrow$ ID) | 1,000 |
| | Test-OOO (OOD $\rightarrow$ OOD $\rightarrow$ OOD) | 1,000 |

### C.2   Training Dynamics and Generalization Patterns

We evaluate model behavior on the 3-hop dataset using the same model, training regime, and diagnostic tools as in the 2-hop setting. Figure 7 summarizes the accuracy trajectories across all test query types and clustering metrics of intermediate representations.

Compared to 2-hop queries, 3-hop reasoning introduces an additional intermediate entity, resulting in two latent steps: $e_1 \xrightarrow{r_1} e_2 \xrightarrow{r_2} e_3 \xrightarrow{r_3} e_4$. To analyze how the model internally represents these latent entities, we extend our diagnostic metrics accordingly.

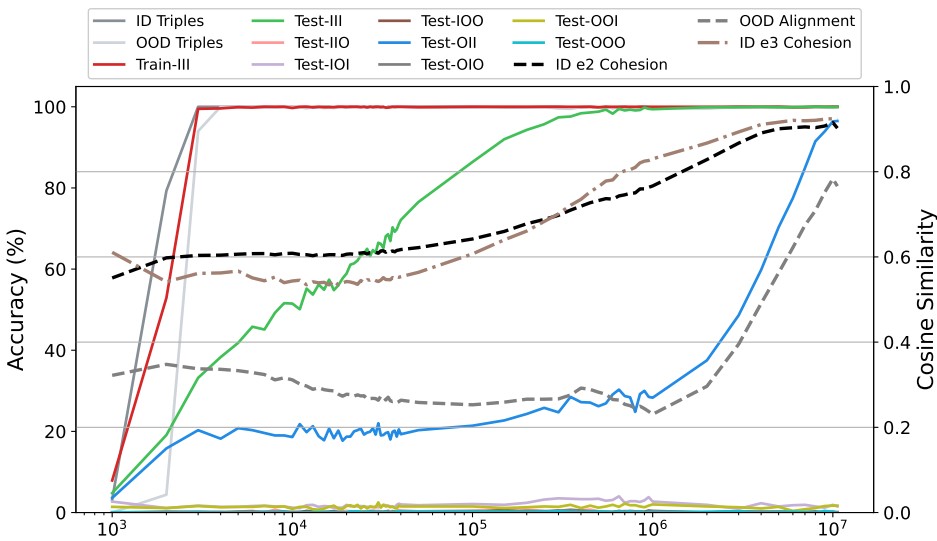

Figure 7: Training dynamics in the 3-hop setting. Accuracy and representation metrics exhibit consistent generalization patterns with the 2-hop case.

Specifically, we extract: (1) $\mathbf{h}_{r_1}^5$, the hidden state at the $r_1$ token in layer 5, to represent the internal encoding of $e_2$ (same as in the 2-hop setting); (2) $\mathbf{h}_{r_2}^6$, the hidden state at the $r_2$ token in layer 6, to represent the encoding of $e_3$.

Based on these, we compute the following representational clustering metrics: (1) ID $e_2$ Cohesion: average cosine similarity among representations of the same ID-derived $e_2$ entity; (2) ID $e_3$ Cohesion: the analogous metric computed over $e_3$ representations. (3) OOD $e_2$ Alignment: cosine similarity between OOD-derived $e_2$ representations and the corresponding ID-derived centroid. We do not compute alignment for $e_3$ because only the first hop ($e_2$) can successfully generalize from OOD inputs.

We observe two key generalization patterns consistent with the 2-hop results:

**(1) In-distribution generalization emerges reliably:** The model successfully generalizes to Test-III queries (ID $\rightarrow$ ID $\rightarrow$ ID), with accuracy rising steadily during training.

**(2) Out-of-distribution generalization remains constrained to the first hop:** Among all OOD-containing query types, only Test-OII (OOD $\rightarrow$ ID $\rightarrow$ ID) shows significant improvement. Other configurations where OOD triples appear in the second or third hop (*e.g.*, Test-IIO, IOI, OOO) fail to generalize. This reinforces the bottleneck observed in the 2-hop case: query-level exposure is necessary for downstream relational generalization.

Furthermore, OOD $e_2$ Alignment closely tracks Test-OII performance, while ID $e_2$ and $e_3$ Cohesion metrics rise in concert with Test-III accuracy. Together, these results highlight that representational clustering at multiple relational depths serves as the internal mechanism enabling successful multi-hop reasoning.

While we do not repeat all behavioral ablations from the 2-hop setting (*e.g.*, acceleration from ID triple exposure, second-hop query-level matching), we note that these phenomena are well explained by the clustering dynamics reported above. In particular, similar representational bottlenecks and alignment requirements arise at each reasoning step, such that second- and third-hop generalization depend on the same clustering dynamics as the first hop. As a result, the cohesion and alignment metrics we report suffice to capture the core generalization behaviors in the 3-hop case.

# D  Additional Validation of Query-Level Requirements in Second-Hop Generalization

## D.1  Verifying Second-Hop Failure under Full Supervision

To ensure the observed failure of second-hop generalization is not an artifact of minimal training configurations, we repeat the second-hop ablation experiment under the full base setting described in Section 2.1. Specifically, we exclude a subset of atomic triples (*e.g.*, $(e_B, r_5, e_F)$) from appearing in any second-hop positions during training, while allowing them in atomic queries or first-hop usage (Figure 8).

Despite the model being exposed to these triples in other structural roles, it fails to generalize to Test-II queries that require them as second-hop compositions. Importantly, performance on all other query types remains unaffected. This confirms that **second-hop generalization requires direct query-level supervision**: exposure to a fact in other contexts is insufficient for enabling its role in compositional reasoning.

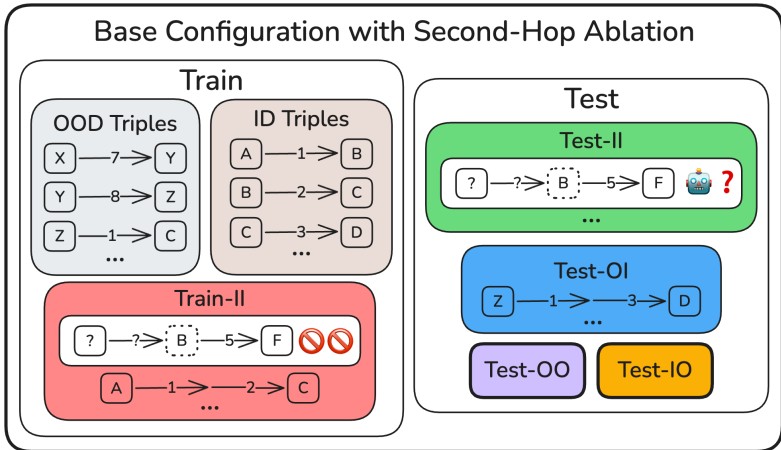

Figure 8: Illustration of the second-hop ablation under the full base configuration. A subset of atomic triples (*e.g.*, $(e_B, r_5, e_F)$) is excluded from appearing as second hops during training. The rest of the data remains unchanged, allowing controlled evaluation of second-hop generalization.

## D.2  Accuracy vs. Second-Hop Exposure Frequency

To further investigate the role of query-level exposure in second-hop generalization, we examine whether second-hop triples that appear more frequently in training queries are learned earlier, using the full base configuration.

We select a fixed training checkpoint in the early part of Phase II—specifically when Test-OI accuracy reaches approximately 50%—and we group second-hop triples by their frequency of occurrence in Train-II queries. For each frequency group $k$, we compute the average accuracy over all Test-II queries whose second-hop triple appeared exactly $k$ times in the training set.

The results reveal a clear trend: higher second-hop exposure frequency during training leads to greater accuracy on corresponding test queries at this intermediate phase (see Figure 9), reinforcing the causal role of second-hop participation in enabling generalization.

# E  Probing Preference for Explicit Decoding

To test whether the model prefers to encode intermediate entities in an explicitly decodable form, we construct a new configuration that reveals the model's decoding preference behaviorally.

This configuration (Figure 10a) removes all ID triples from the training data, but retains (i) all Train-II 2-hop queries, which require reasoning through ID-derived intermediate entities, and (ii) all OOD

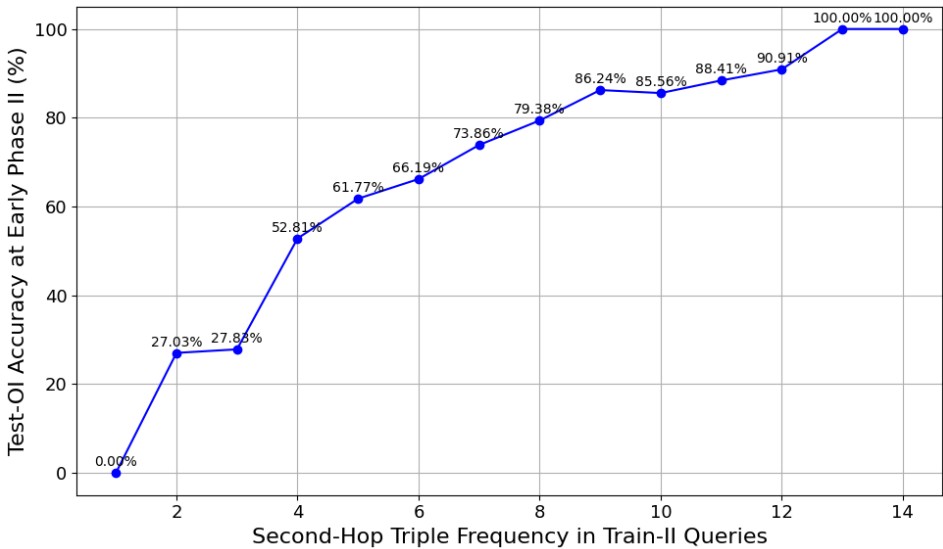

Figure 9: Average Test-OI accuracy at early Phase II (approx. 50%) grouped by second-hop triple frequency in Train-II queries.

triples, which still provide supervision for decoding at the $r_1$ position. Crucially, tail entities in both ID and OOD triples **share the same tail entity vocabulary**, encouraging the model to apply similar decoding strategies across domains.

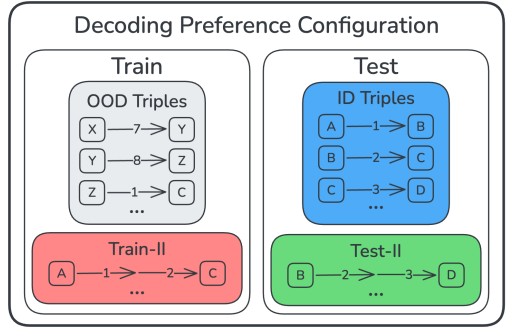

(a) Decoding Preference Configuration

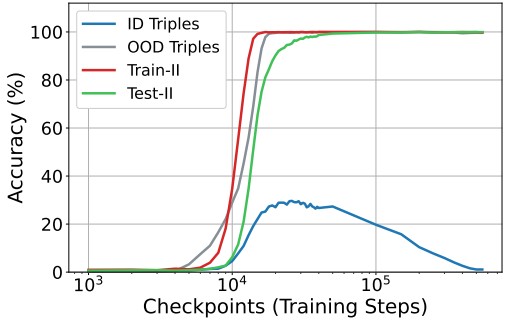

(b) Accuracy dynamics under this configuration.

Figure 10: **Decoding Preference Experiment.** (a) Experimental setup where ID triples are excluded from training and only used for testing., while the model is trained on OOD triples which share the same tail entity vocabulary as the ID triples, and all Train-II queries. (b) Accuracy over training steps shows that the model initially recovers held-out ID triples, suggesting an attempt at explicit decoding, but later abandons this strategy.

This setup leverages the shared hidden state structure between atomic and two-hop queries discussed in Section 4.2, implying that if the model encodes the intermediate entity in a decodable form during 2-hop reasoning, it should be able to recover ID triples (*e.g.*, given a ID query $(e_A, r_1)$, predict $e_B$) even if these triples were never seen during training.

Figure 10b shows the result. Initially, the model answers some ID triples correctly, suggesting an early attempt for explicit decoding. However, this accuracy soon declines, while the performance on Test-II continues to rise. This indicates that the model abandons decodable representations in favor of internal ones that support reasoning but cannot be directly decoded. Explicit decoding, while initially attempted, is not sustained—suggesting it is not the model's preferred solution when alternatives are available.

# F    Additional Results for Cross-Query Causal Patching

We provide the individual patching success rates for each of the three settings: Phase II (ID-derived), Phase III (ID-derived), and Phase III (OOD-derived). These results complement the averaged trend shown in the main text (Figure 4) and confirm the consistency of intermediate entity localization across training stages and generalization regimes.

To provide a full developmental picture, we also include Phase I patching results in Figure 11a. These results exhibit near-zero success across all positions and layers, consistent with the absence of any generalization behavior at this stage. This reinforces our claim that meaningful intermediate representations emerge only after compositional reasoning capabilities begin to develop.

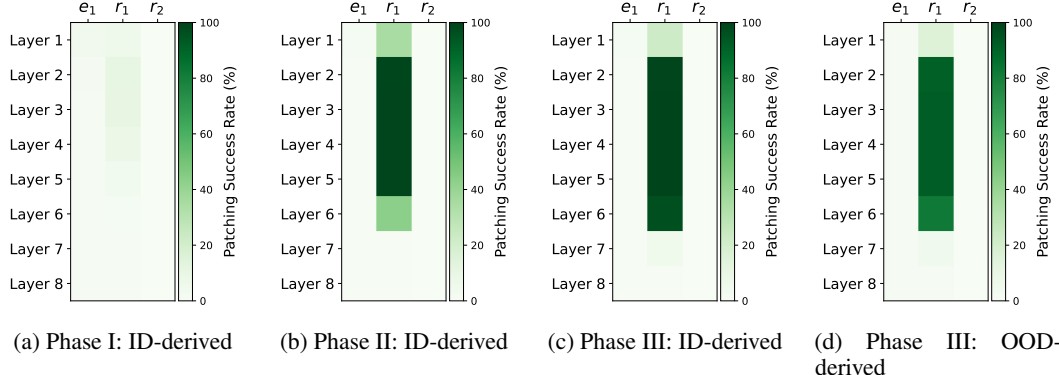

(a) Phase I: ID-derived    (b) Phase II: ID-derived    (c) Phase III: ID-derived    (d) Phase III: OOD-derived

Figure 11: Patching success rate across layers and token positions for different phases and data sources.

# G    Training Details and Bigger Models

## G.1    Training Details

Our training procedure largely follows the public implementation provided by Wang et al. [25], with a few modifications to accommodate our specific experimental setting.

The model is a decoder-only Transformer, identical in architecture to GPT-2, with 8 layers, 768 hidden dimensions, and 12 attention heads. Optimization is performed using AdamW with a learning rate of $1 \times 10^{-4}$, 2000 warm-up steps, weight decay of 0.1, and a batch size of 1024. All models are trained significantly beyond convergence to allow observation of late-stage generalization behavior (Section 2.2).

Training is conducted on NVIDIA RTX 3090 GPUs, and the maximum training duration is extended to 3 weeks to ensure stable cross-distributions generalization. All experiments are implemented using the same PyTorch and Huggingface Transformers framework as in the original codebase.

## G.2    Scaling Analysis: Dynamics and Alignment in Larger Models

We extend our main experimental setup by training a larger model, Qwen2.5-1.5B, under the same base configuration. Our goal is to examine whether the developmental trajectory of multi-hop reasoning observed in smaller models persists at scale, and to further investigate the alignment between behavioral accuracy and representational metrics.

The training progresses successfully through Phase I (memorization) and Phase II (in-distribution generalization), and reaches Phase III (cross-distribution generalization). However, we observe increased instability during Phase III: although the model demonstrates the ability to generalize to Test-OI queries, the performance exhibits significant fluctuations. We report the Test-II and Test-OI accuracies, alongside the ID Cohesion and OOD Alignment metrics (Figure 12).

Interestingly, we find that the Test-II accuracy does not rise in lockstep with the ID Cohesion metric, in contrast to the strong correlation observed in smaller models (Figure 2). We interpret this

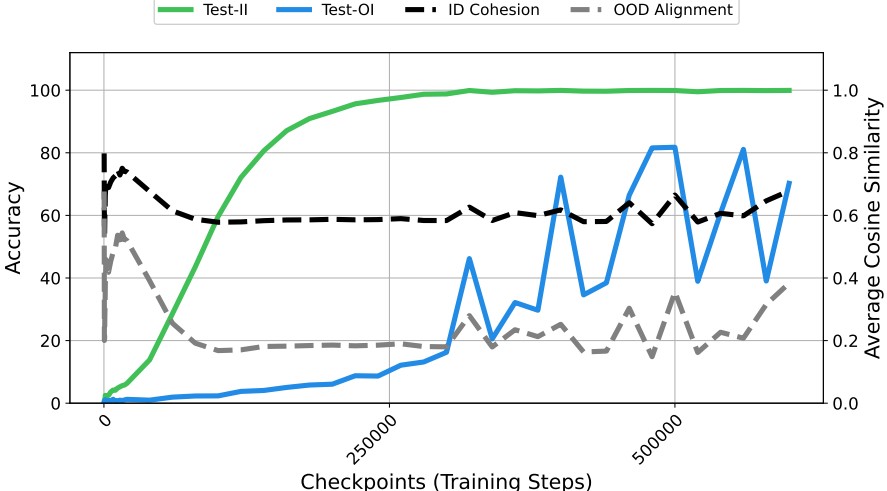

Figure 12: Test-II and Test-OI accuracy and ID Cohesion and OOD Alignment metrics over training steps in the Qwen2.5-1.5B model. Although the model reaches Phase III generalization, substantial variance is observed in both Test-OI accuracy and OOD Alignment, which nonetheless remain tightly coupled. In contrast, Test-II accuracy does not closely track the ID Cohesion metric, suggesting a representational bottleneck at the second relational step.

decoupling as evidence that ID Cohesion is a necessary but not sufficient condition for successful Test-II generalization. While a coherent latent space is required to support compositional reasoning, achieving high Test-II accuracy also depends on the model's ability to utilize these representations in **executing the second relational step**. In other words, beyond aligning representations, the model must also learn to map from an aligned intermediate state to the correct final output via the second-hop relation.

In smaller models, these two aspects—representation alignment and second-hop reasoning—tend to emerge together as part of a single learning phase, leading to tight coupling between ID Cohesion and Test-II performance. In contrast, larger models appear to decouple these processes: representational clustering may occur early, while second-hop reasoning capabilities require additional training to fully mature. As a result, the second relational step becomes the **dominant bottleneck** in Phase II generalization.

This interpretation is further supported by the close alignment between the OOD Alignment metric and Test-OI accuracy. Because second-hop reasoning over ID triples is already well established by Phase II, generalization on Test-OI becomes predominantly constrained by whether OOD-derived intermediate representations have successfully aligned with the ID-centric latent space. This tight correlation holds across model scales: in both the main 2-hop setup with smaller models and the 3-hop results in Appendix C (*e.g.*, alignment between Test-OII and OOD-derived clustering), the emergence of Phase III generalization closely tracks OOD Alignment. In our large-model experiment, although the Test-OI accuracy exhibits high variance, its fluctuations are closely mirrored by the OOD Alignment metric, reinforcing our hypothesis.

We leave the optimization of Phase III training strategies for larger models to future work. Our findings suggest that alignment-based representational diagnostics may serve as useful guides for tuning training schedules or data exposure in this regime, and we encourage future work to explore these directions further.

## H  Validation of Phase III Emergence via ID/OOD Ratio Ablation

To validate our hypothesis in Section 4.1 that the emergence of cross-distribution generalization (Phase III) depends on the dominance of in-distribution (ID) supervision, we conduct an ablation study by varying the ID/OOD ratio of atomic triples under the base configuration.

**Experimental Setup.** We begin with the full base configuration, which includes all atomic triples (both ID and OOD) and the complete set of Train-II queries. We fix the total number of atomic triples and vary the ID/OOD ratio while keeping all other components of training unchanged. Specifically, we test three settings: 80% ID / 20% OOD, 50% ID / 50% OOD, and 30% ID / 70% OOD. In all cases, the **Train-II / ID ratio** is held constant, as prior work Wang et al. [25] identifies this as a critical factor for Phase II generalization.

**Results.** All three ID/OOD configurations unsurprisingly reach Phase I and Phase II, to focus on the emergence of Phase III, we report the Test-OI accuracy and OOD Alignment Score, which capture the model's ability to reason across distributions and align OOD-derived intermediate representations with the ID-induced cluster structure.

As shown in Figure 13, in the 0.8/0.2 setting, both Test-OI accuracy and OOD Alignment Score increase together during training, indicating that the model successfully assimilates OOD-derived intermediate representations into the ID-induced subspace and is able to reuse them for cross-distribution reasoning. In contrast, in the 0.5/0.5 and 0.3/0.7 settings, both metrics remain consistently low, suggesting that the model fails to form aligned representations for OOD triples and consequently cannot generalize to Test-OI queries. This divergence across configurations highlights that strong ID supervision is essential for enabling Phase III generalization.

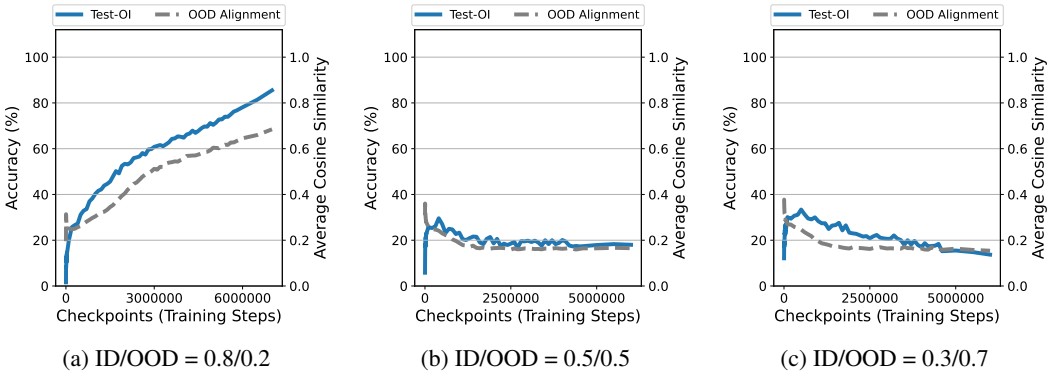

(a) ID/OOD = 0.8/0.2      (b) ID/OOD = 0.5/0.5      (c) ID/OOD = 0.3/0.7

Figure 13: **Test-OI accuracy and OOD Alignment Score** under different ID/OOD splits. All configurations successfully reach Phase I and Phase II. Only the 0.8/0.2 setting supports Phase III generalization, as indicated by joint increases in Test-OI accuracy and OOD alignment. Lower-ID settings fail to align OOD-derived bridge entities, preventing cross-distribution reasoning.

# I   Evidence for Representation Clustering from the Decodable Subspace

To validate the representational mechanism discussed in Section 4.2 that ID triple supervision constrains the $r_1$ representation to lie in a decodable subspace, we design an ablation experiment to test whether held-out ID triples can be recovered solely through shared compositional contexts in Train-II queries.

Specially, We randomly select a subset of ID triples (e.g., $(e_A, r_1) \rightarrow e_B$) to exclude from the atomic triple training set. These held-out triples are removed from all atomic query contexts but remained involved in Train-II queries (*e.g.*, $(e_A, r_1, r_2) \rightarrow e_C$, where $e_B$ serves as the intermediate entity). Crucially, the model retains exposure to other atomic triples that sharing the same tail entity, such as $(e_X, r_7) \rightarrow e_B$, which appear in both atomic and corresponding compositional queries (*e.g.*, $(e_X, r_7, r_3) \rightarrow e_Y$), Figure 14a illustrates the configuration. This configuration enables us to validate whether ID atomic triples supervision constrains the $r_1$ hidden state to a decodable region by testing whether these held-out triples could be recovered.

As illustrated in the Figure 14b, The model successfully recovers held-out triples despite their absence from atomic training. Crucially, this recovery capability emerges concurrently with Test-II generalization, confirming that the model leverages the same intermediate representation subspace for both atomic and compositional reasoning. The results indicate that ID triple supervision accelerates generalization not by providing explicit factual memorization, but by structurally constraining the model's representational space to align atomic and compositional reasoning pathways.

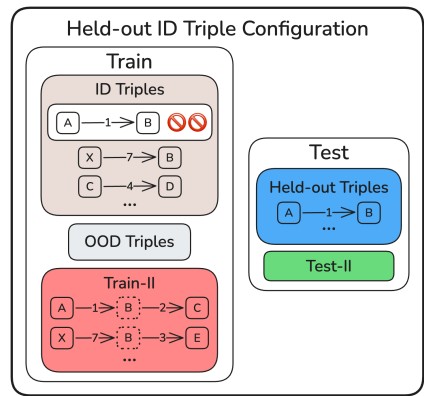

(a) Illustration of the data construction.

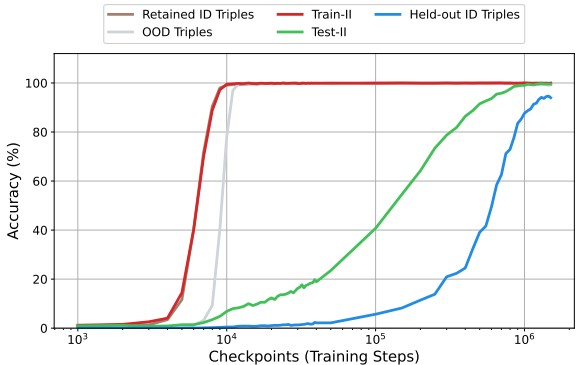

(b) Training curve showing accurate prediction of the held-out triples.

Figure 14: **Verification of ID triple constraint effect.** The model successfully recovers held-out ID triples by leveraging representational constraints from multi-hop supervision and structurally related retained triples, supporting the claim that ID triples accelerate generalization by shaping a decodable representational subspace.

## J   Removing ID Triples Breaks First-Hop OOD Generalization

To test whether ID supervision is necessary for cross-distribution (Test-OI) generalization, we constructed a simplified configuration that removes all ID triples from training. The model is trained only on OOD atomic triples and Train-II 2-hop queries, as illustrated in Figure 15a.

Despite having access to OOD facts and multi-hop supervision, the model fails to generalize to Test-OI queries where the first hop is from an OOD triple. As shown in Figure 15b, Test-OI accuracy remains near chance throughout training. This validates our claim in Section 4.3: without representational anchoring from ID triples, OOD-derived entities cannot support implicit multi-hop reasoning.

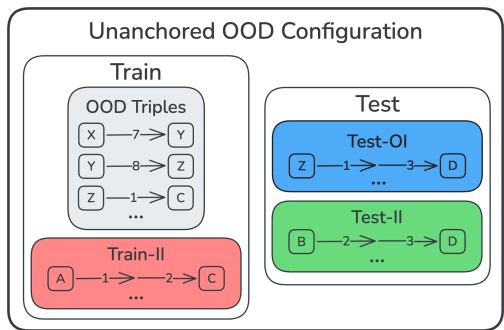

(a) Illustration of the Unanchored OOD Configuration.

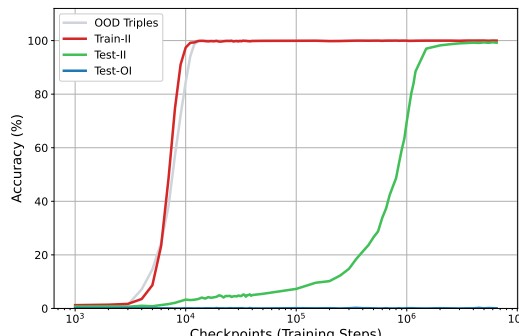

(b) Training dynamics: Test-OI accuracy fails to improve

Figure 15: Validation experiment under ID-removed configuration. Without ID triples, the model fails to reach Phase III, confirming that representational anchoring from ID supervision is essential for OOD-based generalization.

