# OpenReview forum: "How do Transformers Learn Implicit Reasoning?"
_NeurIPS.cc/2025/Conference — NeurIPS 2025 spotlight_

### Official Review · Reviewer_4Y73 · 2025-06-04

**Clarity:** 3
**Significance:** 3
**Originality:** 3
**Rating:** 5
**Confidence:** 4

**Summary:**

This paper investigates how Transformer models learn implicit reasoning by training them on symbolic "fact" triples: in-distribution (in training and in multihop examples) and out-of-distribution (in training but not in multihop). Memorization of in-distribution triples and multihop appears first during training, then novel in-distribution multihop, then out-of-distribution multihop. They find that entity representations appear in middle layers, and in-distribution entity representations begin to cluster earlier in training, then out-of-distribution entity representations begin to align with the in-distribution representations. The multihop reasoning capability of the model for OOD entities depends on the clustering of the OOD entity representations with the in-distribution entity representations. This suggests that multihop abilities for in-distribution entities drives the multihop abilities for OOD entities.

**Questions:**

1. How does the "Cross-Query Semantic Patching" method differ from causal patching in other work (e.g. https://arxiv.org/pdf/2304.14767)? If there are key differences, this would improve my originality score.
2. In the "full-run" probing, why do we expect the logit lens to extract the original entity after passing the patched representation through all model layers? Might the model layers transform the patched entity into some other target entity if a relation is described in the query?
3. In the abstract, the representation results for multihop reasoning are not made clear. E.g. "which reveals that successful reasoning correlates with the cosine-base clustering in hidden space. This clustering phenomenon in turn provides a coherent explanation for the behavioral dynamics observed across training, linking representational structure to reasoning capability. These findings provide new insights into the interpretability of implicit multi-hop reasoning in LLMs, helping to clarify how complex reasoning processes unfold internally and offering pathways to enhance the transparency of such models" is a bit vague, and does not mention any of the key findings, e.g. that alignment of OOD and in-distribution entity representations drives multihop reasoning abilities for OOD entities. I think these results are quite interesting, and it would be good for the abstract to highlight these contributions!

**Ethical Concerns:**

["NO or VERY MINOR ethics concerns only"]

**Final Justification:**

Maintaining my currently positive score. Specifically:
* The author response addresses my question about the comparison of their method to causal patching; based on this, I have increased my originality rating to 3.
* The author response also addresses my concern about the "narrow cone" phenomenon, demonstrating that the result isn't just a by-product of model anisotropy. However, because my overall score is already 5 (recommending acceptance), I do not feel that this addition raises my score further.

**Limitations:**

yes

**Quality:**

3

**Strengths And Weaknesses:**

Strengths:
1. Overall, the results are quite interesting and could have substantial implications for how Transformer language models learn multihop reasoning.
2. The results in Section 2, showing learning of different types of multihop reasoning are interesting and informative.
3. The "Cross-Query Semantic Patching" methodology appears sound, although it is not entirely clear how it differs from causal patching in previous work (e.g. https://arxiv.org/pdf/2304.14767).

Weaknesses:
1. The dataset (synthetic fact triples) generation process could be described more fully in the main text. The task is quite synthetic, although this makes sense to reduce the complexity of the phenomenon that the paper is studying.
2. The two new approaches ("cross-query semantic patching" and the "cosine-based representational lens") do not seem particularly novel (e.g. see https://arxiv.org/pdf/2304.14767; https://arxiv.org/abs/2403.05440; https://arxiv.org/abs/2205.05092). Framing these as main contributions of the paper (e.g. in the abstract) detract from potentially more novel contributions, such as the implications for model learning of multihop reasoning.
3. The finding that OOD entity representations and in-distribution representations increase in cosine similarity throughout training is not entirely surprising given that language model representations usually concentrate in a narrow cone throughout training (e.g. https://arxiv.org/abs/2401.12143). However, it is interesting that this can be linked to multihop reasoning ability to some degree.

---

> ### Author Rebuttal · Authors · 2025-07-31
>
> We thank the reviewer for the thoughtful feedback and for highlighting the strengths of our work.
> In the following, we address the reviewer’s questions and concerns by grouping related points.
>
> ## [W1] Dataset Description:
> We thank the reviewer for the suggestion. We acknowledge that the main text currently provides only a brief description due to space limitations, and many details were placed in the appendix. In the revised version, we will:
> 1. Move **key details** (e.g., ID/OOD ratios) from the appendix into the main text.
> 2. Add illustrative **examples** to help readers better understand how the symbolic dataset mimics realistic multi-hop reasoning scenarios.
>
> We believe these changes will improve clarity.
>
> ## [Q1] Cross-Query Semantic Patching vs. Standard Causal Patching
> We thank the reviewer for the question regarding the originality of our Cross-Query Semantic Patching (CQSP) method. We clarify the key differences below:
>
> **1. Same general family, but different purpose.**
>
> Our method is indeed inspired by the general causal patching paradigm, but CQSP explicitly targets **semantic reusability** of intermediate entity representations across queries, which is essential for studying multi-hop reasoning.
>
> Standard causal patching primarily asks “does this activation (in target query itself) affect the target output?”, whereas our method asks “does the representation from a source query carry reusable intermediate-entity information that can correctly drive reasoning in a target query?”.
>
> **2. Difference in how source activations are treated.**
>
> In standard causal patching, source activations are often treated as **perturbations** (sometimes even randomized) to measure causal effect on the target prompt.
>
> In contrast, CQSP leverages the source prompt representation as a meaningful semantic candidate, inserted into the target prompt’s context, and we evaluate whether it produces the correct 2-hop answer (see line 191).
>
> **3. Stricter criterion**
>
> Standard causal patching does not have a well-defined “success” output—it can only evaluate the **magnitude of output probability shifts** at the target prompt to estimate causal influence at different positions.
>
> In contrast, CQSP has **a known expected answer** (see line 191), only producing this correct answer counts as success.
> This makes our metric goal-directed and semantically meaningful.
>
>
> **4. Concrete example vs. standard patching.**
>
> In our extended setup of [Wang et al. [24]](https://arxiv.org/pdf/2405.15071), standard causal patching was applied to locate circuits encoding intermediate entities.
> Their approach compared Phase I and Phase II causal graphs and subtracted them, which produced **noisy results** (see Figure 9 in their paper).
>
> By contrast, our CQSP method achieves **nearly 100% success** (Figure 4) in pinpointing the positions that encode the correct intermediate entities, providing a far cleaner and functionally interpretable signal.
>
> In short, CQSP is a cross-query, semantically driven extension of causal patching.
> We hope this clarification addresses the reviewer's concern about our methodological novelty and supports an improved originality assessment.
>
>
> ## [W3] Narrow Cone vs. Entity-Specific Clustering
> We thank the reviewers for raising the concern that the observed OOD–ID cosine alignment might be a byproduct of the global “narrow cone” phenomenon (e.g., 2401.12143), where all representations become more similar during training.
>
> Our key distinction is that the alignment we report is entity-specific and functionally predictive:
> 1. We track hidden states of **specific bridge entities**, not global embeddings.
> 2. Their clustering dynamics are tightly **linked to functional behavior**: ID cohesion rises with Test-II accuracy (Phase II), and OOD alignment rises with Test-OI accuracy (Phase III).
> 3. To further rule out global collapse, we computed the average pairwise cosine similarity among **different intermediate entities** over Phases I–III:
>
> | Phase | ID Cohesion ↑ | OOD Alignment ↑ | Cross-Entity Similarity ↓ |
> |-------|----------------|------------------|-----------------------------|
> | I     | 0.407           | 0.188             | **0.068**                    |
> | II    | 0.765           | 0.115             | **0.011**                    |
> | III   | 0.871           | 0.651             | **0.004**                    |
>
> As shown, cross-entity similarity consistently **decreases**, confirming that the clustering is highly entity-specific rather than a byproduct of general compression.
>
> This result directly **refutes** the “all representations become similar” hypothesis and **strengthens our original claim**:
> functional multihop reasoning emerges when bridge-entity representations form tight, entity-specific clusters, not from global embedding collapse.
>
> We will incorporate the findings into the revised version of the paper, and we are grateful to the reviewer for the suggestion that inspired this analysis.
>
> ## [Q2] Full-Run Probing as Stricter Complement
> We appreciate the reviewer’s question regarding why we expect the logit lens to extract the original entity after passing the patched representation through all model layers.
> Our reasoning is as follows:
>
> **1. Empirical observation of early-layer decodability.**
> When we perform cross-query patching at the r1 token position, we observe that:
> + Patching at very **early layers** (1–2) already leads to the final layer decoding the correct intermediate entity with high success.
> + In contrast, direct immediate decoding of hidden states (without full-run propagation) only becomes stable from **layers 4–5** onward.
>
> **2. Role of later layers.**
> This observation indicates that the decodability of the intermediate entity is already established in early layers, and the downstream layers **refine and stabilize** the decodability.
>
> We therefore introduce full-run probing as a **stricter complement to immediate decoding**, ensuring that representations which cannot be immediately decoded, but already contain sufficient information for correct decoding, are not overlooked.
>
> ## [W2, Q3] Contribution Framing and Highlighting Key Findings
> **1. Adjusting the framing of our analysis methods**
>
> We appreciate the reviewer’s comment regarding the novelty of our analysis tools.
>
> Our intention is **not to claim standalone methodological novelty**.
> These tools were designed primarily to **challenge** the prevailing decodability-based assumption in multi-hop reasoning analysis, which assumes that intermediate entities must be explicitly decodable (Section 3.2), and to enable the **mechanism discovery** that forms the core contribution of our paper.
>
> We will revise the final version to clearly present these tools as supporting analysis, with the core contribution framed as the mechanism discovery.
>
> **2. Highlighting the key OOD-ID alignment finding**
>
> We thank the reviewer for emphasizing that the alignment of OOD and ID entity representations is a particularly interesting and impactful finding, which another reviewer also noted as a central and impactful contribution.
>
> We fully agree and will revise the abstract and contribution statements to make this finding explicit.
>
> We will clarify this motivation for the full-run probing setup in the revised version.

---

> ### Comment · Reviewer_4Y73 · 2025-08-01
>
> Thank you for the responses! The author responses address my questions. I am maintaining my currently positive score -- I would also increase my originality rating to 3, but that does not seem to be an option in the review revision. I left a note on the review.
>
> Update: was able to change the originality rating.

---

> > ### Author Response · Authors · 2025-08-06
> >
> > Thank you for your positive feedback and the updated originality rating.

---

### Official Review · Reviewer_mwv1 · 2025-06-30

**Clarity:** 2
**Significance:** 3
**Originality:** 3
**Rating:** 5
**Confidence:** 4

**Summary:**

This paper studies the dynamics of how Transformers learn to perform implicit compositional reasoning, based on the setups in prior work. Specifically, the authors conduct a more detailed examination on top of previous studies, including 1) a more fine-grained classification of out-of-distribution (OOD) scenarios in terms of the sources of constituting facts, where the authors find novel generalization phenomena; 2) a set of further ablation studies and analysis techniques towards confirming and explaining these phenomena, such as intervention methods that integrate causality and semantics, and clustering/visualization approaches which provides statistics that correlate well with generalization behaviors.

**Questions:**

None

**Ethical Concerns:**

["NO or VERY MINOR ethics concerns only"]

**Final Justification:**

The rebuttal phase satisfactorily addressed my concerns, and I will maintain my original positive assessment of the paper.

**Limitations:**

Yes.

**Quality:**

3

**Strengths And Weaknesses:**

Strengths
- This paper reports a set of novel findings on top of previous work in similar directions, including how Transformers can actually generalize in compositional reasoning under some out-of-distribution scenarios (in particular, cases where the first hop is out-of-distribution and the second hop is in-distribution, or 'test-OI'), and also how ID atomic triples are not strictly required for ID generalization, yet, they accelerate ID generalization and encourage the bridge entity representations under OOD scenarios to converge onto those under ID scenarios. These findings are novel and provide interesting insights into the dynamics of how Transformers learn implicit compositional reasoning.
- The authors conduct a series of ablation studies and analysis on the intermediate representations, which provide further consolidation of the results and support for the explanation of the new observations, including the effect of ID triples. These also point to potential ways of analyzing the geometry of internal representations beyond simple projection/probing as done in prior work, and could inspire future studies in related directions.

Weaknesses
- The overall structure/writing could be improved to focus more on the central findings/hypothesis. Details on some ablation studies that are closely relevant are left in the Appendix, while some sections which are not well-connected to the main thesis are unnecessarily detailed. This makes the overall logic flow a bit scattered. I would suggesting moving some non-critical results to the Appendix in exchange for more concrete ablation studies in the main content. For example, I believe the results in Section 2.4 are not well-connected with the rest, and are also mostly reassurance of prior findings.
- While the experiments and analysis provide decent insights into the roles of ID triples, I believe one critical question of why/how they help shape the OOD bridge entity representations during Phase III is still not fully explained, e.g., what could be the reason that test-OI generalization starts to improve at around 10^6 optimization steps (Figure 2)? It would add more depth to the insights if some experiments could be done to characterize this observation and other factors such as regularizations.

---

> ### Author Rebuttal · Authors · 2025-07-31
>
> We thank the reviewer for the thoughtful feedback and for highlighting the strengths of our work.
> In the following, we address the reviewer’s questions and concerns by grouping related points.
>
> ## [W1] Paper Structure
>
> We understand the reviewer’s concern about the organization of the paper, and we clarify our reasoning below.
>
> ### **1. Importance of Section 2.4**
>
> We would like to clarify the role of Section 2.4 in our study.
> Section 2.3 and 2.4 jointly answer the core question of **what data signals are truly required** for implicit reasoning:
> + Section 2.3 demonstrates that **ID atomic triples are not required** for ID generalization, challenging the prevailing assumption in prior works.
> + Section 2.4 goes one step further, **identifying what the model actually depends on** at the query level, revealing that second-hop generalization requires query-level exposure.
> This extends the understanding of data dependency from a macro level (type of supervision) to a fine-grained query level, which is essential to answering how LLMs learn.
>
> Therefore, Sec. 2.4 is not redundant but a key part of the data-dependency analysis.
>
> ### **2. Multi-faceted Structure is Intentional**
>
> Our study is organized to systematically answer “how LLMs learn implicit reasoning” from **multiple complementary perspectives**:
> 1. the role of data (Sec. 2.3–2.4),
> 2. training dynamics and observed phenomena (Sec. 2.2–2.4),
> 3. analysis tools (Sec. 3), and
> 4. internal mechanisms (Sec. 4).
>
> This design is deliberate: phenomena alone cannot explain how learning occurs, and mechanistic insights require the context of observed phenomena.
>
> We also note that different reviewers valued different aspects:
>
> + Both reviewers **mwv1** and **4Y73** found the new phenomena observed in Section 2 interesting and informative.
> + Reviewer **1ywD** highlighted that “The interpretability portion is a highlight.”;
> Reviewer **mwv1** also praised our mechanistic analyses.
> + Reviewer **4Y73** expressed interest in the patching tools.
>
> Together, this indicates that the current structure naturally supports a comprehensive answer to the central question.
>
> ### **3. Open to Further Discussion on Presentation**
>
> We acknowledge the reviewer’s concern about the placement of certain results between the main text and Appendix.
>
> We are **happy to further adjust the presentation**, whether by bringing key results into the main text or by moving less critical details to the Appendix, if the reviewer feels this would improve clarity, and we are open to continued discussion to make the structure clearer for the final version.
>
> ## [W2] Deeper Mechanism Behind Test-OI Generalization
> We appreciate the reviewer’s insightful question on the deeper cause of Test‑OI generalization.
>
> Our study is scoped around how LLMs learn implicit reasoning, and we provide an empirical mechanistic explanation—as also noted by the reviewer in the strength section:
> > "...ID atomic triples...encourage the bridge entity representations under OOD scenarios to converge onto those under ID scenarios.."
>
> We intentionally stopped at this level of analysis for two reasons:
> 1. **Scope alignment with the main research question.**
> Our goal is to characterize how implicit reasoning emerges through data, training dynamics, and internal representations.
> As Test‑OI generalization constitutes only one component of the broader question, our current explanation is both intuitive and sufficient for supporting this specific behavior, without over-extending the paper into a deeper causal analysis.
> 2. **Ensuring rigor and generality.**
> We believe that a rigorous investigation requires a more general study (e.g., connecting to representational collapse), **which cannot be convincingly achieved within the current symbolic setup alone.**
> Therefore, we treat this as an exciting direction for future work rather than something we aim to fully resolve in the current paper.
>
> We will clarify this scope in the revised version and leave a deeper investigation of the timing and underlying dynamics of Test‑OI generalization as a natural direction for future work.
>
> We thank the reviewer again for thoughtful feedback and hope our responses have addressed the concerns raised.

---

> > ### Comment · Reviewer_mwv1 · 2025-08-05
> >
> > Thank you for the detailed and thoughtful rebuttal.
> >
> > Regarding Section 2.4, I acknowledge the authors' clarification of its intended role in connecting query-level exposure to generalization. I still find that its results overlap somewhat with prior work and are less novel than, say, Section 2.3, but nevertheless this is a relatively minor point and does not detract from the overall contribution of the paper.
> >
> > The other concerns I raised have been satisfactorily addressed in the rebuttal. Overall, I maintain my original positive assessment of the paper.

---

> > > ### Author Response · Authors · 2025-08-06
> > >
> > > Thank you for your follow-up and for maintaining your positive assessment.
> > > We greatly appreciate your constructive feedback and and are glad that our clarifications have satisfactorily addressed your concerns.

---

### Official Review · Reviewer_XYMx · 2025-07-02

**Clarity:** 3
**Significance:** 3
**Originality:** 3
**Rating:** 5
**Confidence:** 4

**Summary:**

The paper investigates how LLMs execute multi-hop relational reasoning; specifically, the authors focus on studying the properties of the intermediate/implicit entity representation. The authors construct an artificial dataset and train their model from scratch. They conduct a number of probing and intervention (patching) techniques, obtaining a number of insightful observations.

**Questions:**

## Questions/Suggestions:

### Deliberation tokens
One experiment that'd be very natural in this setting is with using the deliberation or planning tokens (https://arxiv.org/html/2310.05707v4). I.e. if we offer the model an extra token to "think" for longer, it might use this extra token to come up with the implicit entity representation.
For example, the dataset for such an extra experiment could take the form of e1, r1, <deliberation1> r2 <deliberation2> answer.

To clarify: I am not demanding that the authors should do this. Nevertheless, I think such an experiment would substantially strengthen the paper. Especially since in practice, chain-of-thought-like applications are common; so LLMs can implicitly reason not only layer-by-layer, but also by using intermediate tokens to refine their representations.

### Re: clustering:
Figure 5 is generally good and very interesting, but I suggest changing the color of OOD derived entities. The pale-yellow points are extremely hard to see. I didn't realize they were there at all before I read the legend.

Also, in this particular case I feel it'd be good to display an extra control sample: perhaps the entity embedding for the first, explicitly given entity? This is just to better understand if, maybe, all representations are shrinking/aligning in general.
The same applies to the experiment in 4.1. Are the clustering results specific to the "implicit" entity? Alternatively, it could be that any class of stimuli the model is regularly exposed to during training will see this clustering behavior.

### Clarity

Although the paper is generally well-written, at times, the language is very heavy and difficult to parse.
For example, consider this sentence:
"Cross-query semantic patching, which enhances causal interpretability by locating intermediate entity representations based on their semantic transferability across queries rather than solely their impact on final outputs."
For me personally, it was extremely hard to understand what this means. The sentence is very long, and it also uses a lot of terms that were not yet clearly introduced in the paper.
I think it'd greatly benefit the paper if the authors go through it again, rephrasing and simplifying such overly long sentences.

I also strongly suggest reworking the definition of Cross-Query Semantic Patching. Right now the phrasing is confusing. The part "we extract the hidden representation believed to encode the bridge entity" suggests that the authors already know which hidden representation is encoding the bridge entity. Only later the authors clarify that they are trying a number of different options.

I suggest phrasing the approach in a more sequential manner. I.e. the order should be:
1) we want to find which hidden representations encode...
2) for each potential representation, we do ...

It doesn't actually need to be a list, but the order is important. Otherwise, it reads a little backwards and can be very confusing to the reader.

## Minor:
I initially read "We distinguish: Train-II" as "Train 2". It might be good to start with Train OI, for clarity.

**Ethical Concerns:**

["NO or VERY MINOR ethics concerns only"]

**Final Justification:**

After considering the author's rebuttal and other reviews, I decided to increase my initial score by one point.

While some of the issues remain, they are largely in the "desiderata" space. E.g. my suggestion about deliberation tokens. I do think some experiments on them could be added quite easily and would improve the paper, but it's also acceptable to just acknowledge this in the limitation section; I trust the authors will do that.

All in all, I feel that "borderline accept" doesn't do this paper justice, so I have increased my score by one point, to "5, Accept".

**Limitations:**

The limitations are briefly but adequately discussed in the discussion section.

**Paper Formatting Concerns:**

No concerns.

**Quality:**

4

**Strengths And Weaknesses:**

# Strengths:
- The paper addresses a highly relevant problem
- The experimental design and execution are solid
- The results regarding the training dynamics split into different phases are quite insightful.

# Weaknesses:
## Clarity:
- The paper is generally well-organized, but I believe it'd be beneficial to have an extra round of editing, focusing on 1) simplifying/breaking up overly long sentences. Making sure that, on paragraph level, new concepts are introduced clearly before their usage.
It's especially important for everything on Page 2. I believe that the authors are trying to precisely define everything right there, but it's impossible to do so. The resulting definitions are too condensed and abstract to be useful.

When these newly introduced terms are re-used again, it only creates more confusion.
E.g. line 58: "In Section 3, we first use cross-query semantic patching to localize intermediate entity representations, typically identifying them within the middle layers corresponding to the r1 tokens." At this point, the reader only briefly saw one mention of the "r1" tokens, many paragraphs earlier. The definition of "cross-query semantic patching" was given, but it was too abstract to understand what exactly the "cross-query" method does. So it might be better to just drop these specifics, and give a general idea of Section 3. Something like:
"In Section 3, we show that intermediate entity representations typically occur in the middle layers." There will be time to introduce the "cross-query" method later, as well as give more details.
If it is crucial to mention which tokens are involved, something like this could work to remind the reader of the problem structure: "In Section 3, we show that intermediate entity representations typically occur in the middle layers and correspond to \\( r_1 \\) tokens in \\( e_1, r_1, r_2 \\) prompts."

Overall, this weakness is not disqualifying, but I believe that an extra round of editing will greatly enhance the reader experience and, therefore, make the paper more impactful.

## Experimental support

Generally, the experimental support is quite strong. There are a few extra controls that would be beneficial to add in section 4.

Additionally, it seems that a natural thing to look at would be to study whether/how intermediate "deliberation" tokens can be used to "store" the implicit entities. It's unfortunate that the authors did not look into it, because in practice, Chain of Thought - like methods allow the model plenty of room to come up with intermediate computations not only through explicit chain of thought reasoning, but also by using the intermediate tokens to refine their implicit representations.

I give more detail on this in the Questions section.

## Applicability

As any study with artificial data, this one also suffers from a decreased practical applicability. It's not clear whether and how well the results would generalize to LLMs "in the wild." I don't think this is a disqualifying issue, but it would, of course, greatly benefit the paper if the authors included some experiments on, for example, pre-trained LLMs, but with new entities, etc.

The deliberation token experiment I mentioned above would also, indirectly, connect the paper closer to practical applications.

## Exploratory-ness

While the results are valuable, I believe it'd be good to spend more time to integrate them into the existing literature. I.e. the obtained conclusions are sound and interesting, but I sometimes felt that the authors were pursuing interesting results they stumbled upon by chance, rather than testing hypotheses that were formed based on previous research. This is, again, not disqualifying, but it limits the impact of the paper, as it's not clear how to integrate the obtained knowledge into the existing research.

I.e. the authors obtained many interesting observations, but it's not 100% clear how impactful they are and whether they substantially change our understanding of LLM dynamics.

# Conclusion
Generally, the paper is quite strong as is. Although I listed more weaknesses than strengths, some of these weaknesses are essentially different sides of the same issue, and none of them are disqualifying. Overall, I think the paper is sound enough and interesting enough to benefit the community.

---

> ### Author Rebuttal · Authors · 2025-07-31
>
> We sincerely thank the reviewer for the thorough reading of our paper and for the detailed and constructive feedback.
> In the following, we address the reviewer’s questions and concerns by grouping related points.
>
> ## [W2, Q1] Deliberation tokens
> We thank the reviewer for the insightful suggestion regarding the use of deliberation tokens.
> Our goal in this paper is to investigate the emergence of implicit multi-hop reasoning under the maximum architectural constraint—the transformer’s ability to reason without any explicit space or token dedicated to intermediate computation.
> We therefore focus on the standard autoregressive setting, where the model must internally encode and reuse intermediate entities.
>
> That said, we agree that introducing deliberation tokens, as proposed by the reviewer, opens up a natural and practical extension to our study.
> It could allow the model to externalize intermediate representations, which would bridge implicit reasoning and practical CoT-style explicit processes.
> **We will explicitly incorporate this direction into our discussion of future work in the paper.**
>
> ## [Q2] Clustering Generality
> We thank the reviewer for pointing out both the visualization and interpretability issues regarding clustering.
>
> We agree that the pale-yellow in Figure 5 is difficult to distinguish.
> We have already **updated the figure** with more saturated colors, and slightly increased the marker size for better visibility.
> These changes will be reflected in the revised version.
>
> Regarding whether the observed clustering is a global effect, we appreciate the reviewer’s insightful question.
> To address this, we randomly selected a set of different intermediate entities and computed the average pairwise cosine similarity among their representations over training.
>
> We report the average similarity across Phases I–III in the following table:
> | Phase | ID Cohesion ↑ | OOD Alignment ↑ | Cross-Entity Similarity ↓ |
> |-------|----------------|------------------|-----------------------------|
> | I     | 0.407           | 0.188             | **0.068**                    |
> | II    | 0.765           | 0.115             | **0.011**                    |
> | III   | 0.871           | 0.651             | **0.004**                    |
>
> As shown, the cross-entity similarity consistently decreases.
> This trend strongly argues against a global representational collapse and confirms that the observed clustering is **entity-specific**, not a byproduct of general compression or training dynamics.
> We hypothesize that this decreasing trend results from entity representations being actively **pulled toward their own ID-derived cluster centers**, which increases intra-entity similarity while decreasing cross-entity overlap.
>
> **This additional result provides even stronger support for our original conclusion.**
> We will incorporate the findings into the revised version of the paper, and we are grateful to the reviewer for the suggestion that inspired this analysis.
>
> ## [W3] Applicability
> We thank the reviewer for raising this important point regarding the practical applicability of our findings.
> While our main experiments are conducted in a controlled symbolic environment, we believe our conclusions offer useful insights into how implicit multi-hop reasoning may operate within pre-trained LLMs as well.
>
> **Regarding applicability to LLMs "in the wild":**
>
> Although we do not directly evaluate pre-trained models, **our findings align well with several recent empirical results on LLMs and provide coherent explanations for their observed behaviors**:
> 1. Our finding in Section 2.3—that models don't rely on atomic triples to learn implicit reasoning—helps explain why single-hop knowledge editing fails to transfer to multi-hop queries ([Zhong et al., 2023](https://arxiv.org/pdf/2305.14795)).
> Conversely, Section 2.4 shows that such reasoning is acquired from compositional data, which aligns with recent findings that editing LLMs using multi-hop prompts leads to better generalization on multi-hop tests ([Yao et al., 2025](https://arxiv.org/pdf/2503.16356), [Zhang et al., 2024](https://www.arxiv.org/pdf/2410.06331)).
> 2. Our finding in Section 2.4 also sheds light on why exposing LLMs to richer compositions improves multi-hop reasoning ([Zhang et al., 2024](https://arxiv.org/pdf/2409.14057), [Xu et al., 2024](https://arxiv.org/pdf/2504.00472)).
> 3. Our Section 4.3 result—that generalization to OOD triples only occurs in the first hop—offers a mechanistic interpretation of the observation in [Feng et al., 2024](https://arxiv.org/pdf/2412.04614), where second-hop OOD queries are significantly more difficult than first-hop OOD ones.
>
> In short, our findings offer a mechanistic lens through which to understand a range of puzzling behaviors already reported in pre-trained LLMs.
>
> **Regarding fine-tuning on pre-trained models with new entities:**
>
> We agree that fine-tuning LLMs on new knowledge could be a path to validate our findings.
> However, multiple recent works have found that knowledge acquired during fine-tuning can **behave quite differently** from knowledge learned during pretraining, and **may even distort pre-existing structure** ([Gekhman et al., 2024](https://arxiv.org/pdf/2405.05904), [Zucchet et al., 2024](https://arxiv.org/pdf/2503.21676)).
> As our primary goal is to understand the mechanism through which implicit multi-hop reasoning emerges, we chose to train models from scratch in a controlled environment.
>
> Nonetheless, we acknowledge that incorporating experiments with pre-trained LLMs and injected new entities would be a valuable future direction, and we are actively exploring such extensions to test the generality of our conclusions.
>
> ## [W4] Clarifying Motivation and Connection to Prior Work
> We thank the reviewer for pointing out that parts of our study may appear exploratory and not fully integrated with prior research.
> We acknowledge that this impression likely arises because our motivation was not stated clearly enough in the current version.
>
> An important motivation is the **lack of consensus in prior works** regarding how LLMs perform implicit multi-hop reasoning and whether they can reliably do so, existing studies report conflicting conclusions on key aspects (see Appendix A).
> As described in line 34 of the paper, we believe that a key reason for this inconsistency is:
> > ...training data lacks precise experimental control, making it challenging to conclusively determine...
>
> To address this gap, we designed a **highly controlled symbolic setup** to systematically revisit these questions.
>
> We appreciate the reviewer’s suggestion to better connect our findings to the prior works.
> In the revised version, we will **make this motivation explicit in revised version and clarify how our study aligns with and challenges prior assumptions**.
>
> ## [W1, Q3] Clarity
> We appreciate the reviewer’s helpful suggestions on improving the clarity and readability of our paper.
> **We will make the following revisions in response:**
> + Simplify and break down long sentences. We will break down long sentences to make the language more concise and direct, while preserving the intended technical content.
> + Terminology Clarification.
> We will ensure that all technical terms (especially on page 2) are clearly defined at their first mention.
> + Rework the definition of Cross-Query Semantic Patching.
> We will present the procedure in a clear and sequential manner, consistent with the reviewer’s suggestion.
> + We will avoid the “Train-II vs. Train 2” ambiguity by first introducing Train-OI as recommended by reviewer.
>
> Again, we sincerely thank the reviewer for the careful reading and thoughtful suggestions.
> We hope that our responses have addressed your concerns and clarified the paper.
> We would be happy to provide any additional clarifications if needed and look forward to further discussion.

---

> > ### Author Response · Authors · 2025-08-08
> >
> > As the discussion phase is approaching its end, we would like to check whether our clarifications and the additional experiments above have addressed the reviewer’s remaining questions.
> > We are happy to provide any further details or clarifications in the remaining time of the discussion phase if the reviewer thinks it would be helpful.
> >
> > We sincerely thank the reviewer for their thoughtful feedback and engagement.

---

### Official Review · Reviewer_1ywD · 2025-07-04

**Clarity:** 2
**Significance:** 3
**Originality:** 2
**Rating:** 4
**Confidence:** 4

**Summary:**

This work studies multi-hop implicit reasoning in Transformer models. The task structure builds on Wang et al. [24]: given a set of entities $e_i$ and relations $r_j$, there exists a set of "atomic facts" $(e_i, r_j) \to e_k$. A 2-hop query takes the form $(e_1,  r_1, r_2) \to e_3$, requiring the atomic facts to be composed: $r_2(r_1(e_1))$. This work studies the ability of Transformers to learn such a task, including the ability to generalize out-of-distribution. The paper begins by presenting an evaluation of in-distribution and out-of-distribution generalization throughout training (*what* the model does), followed by a "interpretability" analysis (*how* the model achieves this behavior internally). Among the results, this work finds that the model first learns to memorizes, then generalizes in-distribution, then generalizes out-of-distribution, in three phases, similar to the grokking behavior described by Wang et al. [24]. In analyzing the internal representations, the authors find that the ability to decode entities from internal representations does not explain when the representation contributes to reasoning. In response, they study the geometry of the model's embedding space using cosine similarity.

**Questions:**

* In Figure 5, why are there so few OOD-derive points? Is the MDS projection computed using only the points displayed in the figures?

Other minor feedback:
* The title sounds a bit grammatically awkward to me. Perhaps "How do Transformers Learn Implicit Reasoning?" would be better.
* When stating “we adopt GPT-2 as our base model,” please clarify that this refers to the GPT-2 architecture, not the pretrained model, and specify the model size (e.g., GPT-2 small) explicitly. This will help avoid confusion.
* Some sentences in the paper would benefit from simplification. Certain phrasings appear overly complex or indirect.

**Ethical Concerns:**

["NO or VERY MINOR ethics concerns only"]

**Final Justification:**

Overall rating maintained, but increased significance rating.

The authors responded to each point raised in the review and clarified their position. These concerns were not fully addressed since they pertain to core aspects of the work, but the authors' response is appreciated and does help communicate their position. The authors promised to address some of the presentation issues, acknowledged overlap in contribution with Wang et al. and clarified where their primary contributions lie, and acknowledged the limitations regarding single-trial experiments.

**Limitations:**

The authors adequately addressed the limitations and potential negative societal impact of their work

**Quality:**

2

**Strengths And Weaknesses:**

**Strengths**
* The interpretability portion (Section 3) is a highlight and offers novel insights. In particular, the observation that entity decodability does not predict reasoning behavior is interesting and important, given the prevalence of decodability-based analysis in recent work. Similarly, the finding of cosine-space clustering aligning with when generalization capabilities emerge is a satisfying insight.
* The work tackles interesting and well-formed questions that are highly relevant for understanding how Transformer models generalize and reason compositionally, which aligns with ongoing interest in grokking and systematic generalization.

**Weaknesses**

* There are some presentation issues, especially in earlier sections of the paper, which can be improved. For example, the introduction uses jargon and terminology (e.g., "ID triples") that are only defined later in the paper. Section 2 appears to be written in a way that assumes strong familiarity with Wang et al. [24], making the paper less self-contained. In particular, it does not spend enough time motivating and clearly defining the task, instead jumping to the description of different generalization splits. I would recommend revising this section to include a motivation and a clearer explanation of the task, including examples.
* Additionally, since this work builds heavily on Wang et al. [24], some aspects of the technical contribution are incremental. For example, the design of the symbolic task underlying this investigation and observations of grokking-like behavior have been studied in previous work.
* Finally, the experiments contain a single trial per configuration. This can be an important limitation for Figures 2 & 3 especially which concern the timing of the emergence of different behaviors, since randomness can play a role here. I would recommend running several trials and including error bars. This seems feasible given the use of relatively small models on synthetic data.

---

> ### Author Rebuttal · Authors · 2025-07-31
>
> We thank the reviewer for the thoughtful feedback and for highlighting the strengths of our work.
> In the following, we address the reviewer’s questions and concerns by grouping related points.
>
> ## [Q1] Justifying the Low OOD-Derived Point Ratio in Figure 5
> We thank the reviewer for pointing this out.
> It is correct that the number of OOD-derived points in Figure 5 is smaller than ID-derived points.
> This stems from our intentional data imbalance.
> As justified in Appendix B (line 543), this **high ID/OOD ratio reflects the realistic distributional skew seen in LLM pretraining corpora**:
> >...they reflect plausible properties of real-world pretraining corpora...most knowledge entities are learned in richly connected contexts (analogous to our ID triples), while only a minority are sparsely anchored or appear in isolation (analogous to our OOD triples)...
>
> This design enables our symbolic environment to approximate realistic structure.
>
> To further address the potential concern about this imbalance, **we also conduct an ablation study in Appendix H**.
> We show that only high ID ratios lead to successful cross-distribution generalization.
>
> ## [W2] Novelty Beyond Wang et al. [24]
> We appreciate the reviewer’s assessment regarding the relationship with Wang et al. [24].
> While our work indeed builds on their symbolic environment, its **novelty lies in substantial extensions in both breadth and depth**, as detailed below.
>
> Our contributions cover four key dimensions of multi-hop implicit reasoning:
> 1. **Data Requirements.** Our findings in Sections 2.3 and 2.4 challenge the prevailing assumption that LLMs perform implicit reasoning by merely combining known atomic triples (see Appendix A).
> We identify what data actually supports generalization, and further uncover fine-grained query-level requirements.
> 2. **Training Dynamics.** We uncover a three-phase progression of learning and explicitly characterize the boundaries of OOD generalization under diverse training configurations, including the 3-hop setting.
> In contrast, Wang et al. [24] concluded that OOD generalization does not emerge.
> 3. **Interpretability Tools.** We introduce new tools to probe internal representations.
> 4. **Mechanistic Insight and Disproof of Prior Assumptions.** As the reviewer noted, our analysis directly challenges the widely held assumption—present in Wang et al. [24] and others—that decodability implies reasoning utility.
> Instead, we demonstrate that successful reasoning aligns with representational clustering in hidden space.
>
> To be transparent, we do overlap with Wang et al. [24] in two respects:
> 1. Dataset Construction. We adopt their symbolic setup, but we **extend it with new training and test configurations** that enable fine-grained ablations.
> 2. Grokking Observations. Our Phase II aligns with their grokking transition. However, while Wang et al.[24] emphasized that grokked models exhibit implicit reasoning capabilities, we focus on **how these capabilities emerge and how they generalize—including to OOD settings**.
>
> **Together, these overlaps represent only a small part of our contribution.** The core of our work centers on fine-grained data requirements, generalization boundaries, and mechanistic interpretability—areas not well-addressed in Wang et al. [24].
>
> ## [W3] Regarding Single-Trial Experiments
> > "the experiments contain a single trial per configuration...I would recommend running several trials and including error bars..."
>
> We appreciate this insightful suggestion.
> Due to the limited time available in the rebuttal period, we are unfortunately unable to rerun full training trials.
>
> That said, we have already conducted experiments under several variant configurations (e.g., with 1000 vs. 2000 entities), which introduce variation in data scale while preserving task structure.
> **These results consistently show the same behavioral trajectory and representational patterns, suggesting that our findings are robust to data-level perturbations.**
>
> We will include these results in the appendix of the revised version as additional support for the stability of our observations.
>
> ## [W1, Q2] Writing and Presentation Improvements
> We appreciate the reviewer’s helpful suggestions on improving the clarity and readability of our paper.
> **We will make the following revisions in response:**
> + Title Revision:
> We agree that "How do Transformers Learn Implicit Reasoning?" is more natural and grammatically correct. We will adopt this revised title.
> + Terminology Clarification:
> We will ensure that all technical terms (e.g., ID triples) are clearly defined at their first mention.
> + Improved Motivation and Task Definition in Section 2:
> We acknowledge that the current version of Section 2 assumes prior familiarity.
> In the revision, we will provide a clearer motivation and explicitly define the reasoning task with illustrative examples.
> + Model Description Clarity:
> We will clarify that “GPT-2” refers to the architecture only and specify the model size explicitly.
> + Simplifying Complex Phrasings:
> We will revise selected portions of the text to make the language more concise and direct, while preserving the intended technical content.
> We agree that some sentences are unnecessarily complex in the current form.
>
> We thank the reviewer again for thoughtful feedback and hope our responses have addressed the concerns raised.

---

> > ### Comment · Reviewer_1ywD · 2025-08-02
> >
> > I'd like to thank the authors for their detailed responses. I will maintain my positive score, leaning towards acceptance.

---

> > > ### Comment · Reviewer_1ywD · 2025-08-05
> > >
> > > Dear authors,
> > >
> > > Thank you again for your rebuttal. Just wanted to send a brief additional note to expand on my feedback and your responses.
> > > - On W1 (presentation): I'm glad to hear that the authors will aim to improve the presentation. As I was reading it, the main obstacle was the use of technical jargon before it was defined (e.g., in the introduction). I would suggest staying at a high-level in the introduction and using more generic language. Similarly, to make the paper more accessible, it should not assume strong familiarity with Wang et al. The other stuff is more minor and aesthetic, but I would recommend refining the writing to make the phrasing simpler and direct (at times, the writing sounded like it was rephrased by an LLM but made to be overly complex or indirect). This is minor.
> > > - On W2 (overlap in contributions with Wang et al): I appreciate the authors' clarifying their contributions in comparison to Wang et al. I recognize that the authors' main contributions lie in aspects such as the mechanistic interpretation, as well as extensions Wang et al.'s set up and findings. I appreciate the authors transparency on the overlap with Wang et al., which lies in the dataset construction and the grokking observation---this is what I am referring to. I recognize that this work makes contributions elsewhere.
> > > - On W3 (single-trial experiments despite a synthetic set up and claims based on the timing of the emergence of different phenomena): I understand that it may not be feasible to run several trials within the rebuttal period. But I still view this as a limitation of the original submission, especially since some of the claims pertain to the *timing* of the emergence of different behavior and since the experimental setup is synthetic of a relatively modest scale. Including results from multiple configurations helps to some degree, and I encourage the authors to include them (especially if there are interesting insights from the differences in configs), but including results with the same configuration and different random seeds would be important for rigor.
> > >
> > > Overall, I will maintain the current overall score of 4, continuing to lean towards acceptance, based on the overall level of contributions made in this work and in light of some of the limitations discussed above. I will increase the "significance" rating from 2 to 3.

---

> > > > ### Author Response · Authors · 2025-08-06
> > > >
> > > > Dear Reviewer,
> > > >
> > > > Thank you very much for your additional follow-up and for recognizing the contributions of our work.
> > > > We truly appreciate your constructive suggestions on improving the presentation, clarifying terminology, and providing additional experimental rigor through multi-trial evaluations.
> > > > We will incorporate your suggestions into the final version.

---

### Note · Authors · 2025-08-12

We would like to highlight the paper’s core strengths and summarize updates from the rebuttal period.

## Core Strengths
1. **Importance.**
This paper addresses an underexplored yet impactful question.
Although current multi-hop reasoning often relies on test-time scaling (e.g., CoT), as model capabilities grow, stronger and more efficient implicit reasoning will be a key improvement direction, making its mechanisms impactful to study.

2. **Novelty.**
Existing work lacks consensus.
We challenge common assumptions—e.g., that models compose atomic facts, that decodability reflects internal reasoning, and that OOD implicit reasoning is impossible—and provide new corresponding explanations.

3. **Completeness.**
The study covers the full process from data and training dynamics to interpretability tools and mechanistic insights.
We first explore the limits of implicit reasoning via diverse data configurations, uncovering new phenomena;
then propose new interpretation methods to analyze internal mechanisms;
and finally use these mechanistic conclusions to explain the observed behaviors.
**Each step yields novel findings and is reinforced by targeted supplementary experiments.**
For example, diverse controlled configurations reveal new behaviors (three-stage learning, query-level supervision requirement), new tools localize and characterize intermediate entities, and geometric analyses explain behavioral dynamics.

4. **Long-term Impact.**
As noted by Reviewer 4Y73, this work “could have substantial implications” for understanding true LLMs;
in discussions with Reviewer XYMx, we further demonstrated how our conclusions can explain phenomena observed in other LLM-based studies of this problem.
What's more, given the prevalence of decodability-based methods in interpretability research, our findings call for re-evaluating their reliability.

## Key Updates from Rebuttal
1. **New metric.**
In response to Reviewers 4Y73 and XYMx, we introduced the cross-entity similarity metric, which strengthens our original findings and will be included in the revised version.
2. **Writing improvements.**
As detailed in the rebuttal, we committed to making the necessary writing refinements in the final version.

All reviewers indicated that their concerns were resolved during the discussion, and we plan to expand certain points with additional discussion in the final version.

---

### Decision · Program_Chairs · 2025-09-17

**Decision:**

Accept (spotlight)

**Comment:**

This paper studies the dynamics of how implicit reasoning successfully arises in language models, based on a symbolic multi-hop task from prior work. All reviewers unanimously appreciated the contributions as significant and interesting, the experiments providing solid support for the claims. Multi-hop reasoning is an important problem to understand -- there's both theoretical and empirical literature pointing out the challenges of multi-hop reasoning; thus it is instructive to see a positive result and a clean analysis of its emergence in a symbolic setting. The community will find the insights here useful.

There are writing issues that multiple reviewers pointed out, which I hope the authors will fix.

Here are a couple of suggestions for pre-2025 related work that would bolster the context given in this paper. [1] present a negative result, whereas [2] present a positive result in path-finding over the weights of a model, which involves a form of implicit reasoning.

- [1] Siwei Wang, Yifei Shen, Shi Feng, Haoran Sun, Shang-Hua Teng, and Wei Chen. ALPINE: unveiling the
planning capability of autoregressive learning in language models. In Advances in Neural Information
Processing Systems 38: Annual Conference on Neural Information Processing Systems 2024, NeurIPS
2024, Vancouver, BC, Canada, December 10 - 15, 2024, 2024.

- [2] Mikail Khona, Maya Okawa, Jan Hula, Rahul Ramesh, Kento Nishi, Robert P. Dick, Ekdeep Singh
Lubana, and Hidenori Tanaka. Towards an understanding of stepwise inference in transformers: A
synthetic graph navigation model. In Forty-first International Conference on Machine Learning,
ICML 2024, Vienna, Austria, July 21-27, 2024. OpenReview.net, 2024.